# Unexpectedly uneven distribution of functional trade-offs explains cranial morphological diversity in carnivores

Gabriele Sansalone [1,2,3] ✉, Stephen Wroe[2], Geoffrey Coates[2], Marie R. G. Attard [2,4] & Carmelo Fruciano [1,5,6] ✉

Functional trade-offs can affect patterns of morphological and ecological evolution as well as the magnitude of morphological changes through evolutionary time. Using morpho-functional landscape modelling on the cranium of 132 carnivore species, we focused on the macroevolutionary effects of the trade-off between bite force and bite velocity. Here, we show that rates of evolution in form (morphology) are decoupled from rates of evolution in function. Further, we found theoretical morphologies optimising for velocity to be more diverse, while a much smaller phenotypic space was occupied by shapes optimising force. This pattern of differential representation of different functions in theoretical morphological space was highly correlated with patterns of actual morphological disparity. We hypothesise that many-to-one mapping of cranium shape on function may prevent the detection of direct relationships between form and function. As comparatively only few morphologies optimise bite force, species optimising this function may be less abundant because they are less likely to evolve. This, in turn, may explain why certain clades are less variable than others. Given the ubiquity of functional trade-offs in biological systems, these patterns may be general and may help to explain the unevenness of morphological and functional diversity across the tree of life.

Phenotypic diversity – and the rate at which it accumulates – is unequally distributed among and within clades, and determining which factors facilitate or limit evolution remains hotly debated[1–4]. Identifying which environmental or organismal factors may either constrain or promote the accumulation of phenotypic variation is therefore central to our understanding of biodiversity[5,6]. Usually, a distinction between extrinsic and intrinsic factors is made, with the former (e.g., new ecological niches becoming available and favouring adaptation) studied and invoked to explain macroevolutionary

patterns of diversity more often than the latter (e.g., pre-existing constraints due to genetic correlations among parts of an organism[7,8]). However, intrinsic and extrinsic factors are likely interacting, with the latter dependent on what is made possible by the former. A special place within this picture is occupied by the relationship between form and function and the existence of functional trade-offs, which arise when a morphological structure has multiple functions[9–13].

Functional trade-offs are inherent to many anatomical structures and are generally considered to act as constraints on phenotypic

[1]Institute for Marine Biological Resources and Biotechnology (CNR-IRBIM), National Research Council, Via S. Raineri 4, 98122 Messina, Italy. [2]Function, Evolution and Anatomy Research Lab, Zoology Division, School of Environmental and Rural Science, University of New England, Armidale, NSW, Australia. [3]Department of Life Sciences, University of Modena and Reggio Emilia, Via Campi 213D, 41125 Modena, Italy. [4]British Antarctic Survey, High Cross, Madingley Road, CB3 0ET Cambridge, UK. [5]National Biodiversity Future Center, Piazza Marina 61, 90133 Palermo, Italy. [6]Department of Biological, Geological and Environmental Sciences, University of Catania, via Androne 81, 95124 Catania, Italy. ✉e-mail: gabriele.sansalone@unimore.it; carmelo.fruciano@unict.it

evolution[14,15]. Trade-offs occur when a trait performs two or more competing functions, making it impossible for selection to simultaneously optimise performance for all functions[16-18]. In other words, morphological structures – and therefore the organisms bearing them – cannot be "good at everything". For this reason, it has been postulated that traits impacted by functional trade-offs should display lower morphological disparity and slower evolutionary rates[14,19,20]. However, there is a growing body of evidence suggesting that stronger performance trade-offs (greatly optimising one one function at the expense of another) are associated with faster evolutionary rates and higher disparity[19,21,22]. The effect of trade-offs on phenotypic evolution – and whether such an effect is consistent across clades and traits – therefore remains unclear, understudied and underappreciated.

Previous investigations of traits involved in trade-offs have either (i) considered a single functional metric (a single metric is used, with extreme values corresponding to optimising one of two functions, which constrains along a single axis the range of possible combinations between the two functions) or (ii) compared, for a given trait, distinct clades that have each optimised one of the possible functions, and tested for differences between clades in evolutionary rates and morphological disparity[14,19,23]. These approaches offer an intuitive way of quantifying the tempo and mode of evolution of the traits subject to a trade-off, e.g., examining the evolution of one function at time. However, treating each function separately (i.e., one function per clade) does not account for the combined effect that these functions exert on anatomical structures. Indeed, the net performance (and consequence on fitness) of distinct and competing functions will be determined by the summed contribution of each function[16,24-27]. That is, examining each function of a given morphological structure separately does not account for the interaction between functions. This is far from trivial, as the interaction between functions and its effect on phenotypic evolution can be complex[24,28]. This issue has spurred the development of a different approach which explicitly models the interaction between functions, a method based on the idea that the relationship between the phenotype and performance for a single function can be represented by a performance surface[29-31]. Performance surfaces for two functions of the same morphological structure can then be combined into a functional adaptive landscape[16]. The functional adaptive landscape is a model of how each phenotype corresponds to a value of the joint performance across the two functions. In this framework, for each phenotype – be it a hypothetical phenotype or the phenotype displayed by a real taxon – one can also estimate the relative weight (importance) of each trading-off function in producing the joint performance[15,16,25,26,32]. Hence, one can quantitatively estimate a quantity – termed "trade-off weight" – which reflects whether a given species remains more "generalised" in the use of a given structure (intermediate values of trade-off weight) or tends to improve one function at the expense of the other.

In an adaptive evolutionary landscape, trade-offs determine the space that forms can explore and can shape the pathways toward adaptation[11,33]. Furthermore, the trade-off between distinct functions can produce combined performance peaks in phenotypic space corresponding to a narrow set of trait combinations (phenotypic disparity), whereas other peaks may correspond to a broader range of phenotypes. Understanding how (mode) and how fast (tempo) species navigate across this space will help determine to what extent trade-offs affect the accumulation of phenotypic and functional diversity across evolutionary time[11,30].

Here, in order to quantify these relationships, we ask two questions: (1) is there a correlation between on one hand, functional trade-offs (i.e., which function is maximised and to what extent) and their rates (i.e., how fast the relative importance of different functions evolves) and on the other hand morphological evolutionary rates? (2) How – if at all – do trade-offs influence the distribution of morphological disparity?

To answer these questions, we focus on the widely recognised force-velocity trade-off[34-37] in mammalian jaws (i.e., the trade-off between how strongly and how quickly a bite can be given). Specifically, we assess the effect of this trade-off on both morphological rates of evolution and disparity of the skull of extinct and modern placental carnivorans and non-herbivorous marsupials. Carnivory evolved multiple times within Metatheria[38], hence comparisons between eutherian carnivorans and metatherian 'marsupicarnivore' taxa (hereafter collectively "carnivores") can also contribute addressing the long-standing question about developmental constraints on marsupial functional diversity[38-42]. We note that both groups contain many species which are more strictly classified as omnivorous.

Mammalian carnivores are an excellent model to study the relationship between form, function, and diversity as they display a wide array of morphological and dietary diversity encompassing different lifestyles[38,43-45]. Carnivores' organismal design is generally thought to reflect resource use and studies using several functional metrics have supported a correlation between diet and bite force[43,45]. The association between bite force and diet is frequently investigated because improving bite performance can allow access to novel ecological niches[43,46]. However, despite this advantage, increasing bite force may come at the expense of jaw-closing speed, reducing the capacity to consume more agile, and typically smaller, more elusive prey[47-49]. Since jaws are fundamental in food apprehension and dispatch, this mechanical trade-off can have far-reaching evolutionary consequences[34,37,50].

In this study, we compared cranial morphology and functional performance of 132 species from 12 families of terrestrial carnivores. Our sampling included members of the families Canidae, Eupleridae, Felidae, Herpestidae, Hyaenidae, Mustelidae, Procyonidae, Ursidae and Viverridae among placentals and Dasyuridae, Myrmecobiidae and Thylacoleonidae among marsupials. We used 3D shape data and geometric morphometrics to quantify carnivore cranial morphology.

For the functional component of this study, we used a theoretical landscape approach[16,26,51] and, through finite element analysis, we estimated force and velocity metrics. We then interpolated the functional surfaces for each function, combined them into a functional landscape, and quantified the relative contribution of each function (trade-off weight) using maximum likelihood[15,16]. We then estimated and compared the evolutionary rates for both cranial morphology and the trade-off between force and velocity. Finally, we used a sliding window approach to assess, at each trade-off weight, whether and to what extent levels of morphological variation in real species (observed disparity) matched the levels of variation across all theoretically possible shapes with that trade-off weight (theoretical disparity). Here, we show that rates of morphological evolution are largely decoupled from evolutionary rates of the change in the relative importance of velocity and force. Further, we show that most of the landscape describing the relationship between form and function was characterised by greater relative importance of velocity, whereas a much smaller phenotypic space was occupied by shapes optimising bite force at the expense of velocity. This pattern of differential representation of different functions in theoretical morphological space is highly correlated with patterns of actual cranial morphological disparity. We do not detect substantial differences between placentals and marsupials, which suggests that similar species across these two distantly related groups have followed similar evolutionary pathways. We hypothesise that many-to-one mapping of cranium shape on function (i.e., distinct shapes producing similar levels of functional performance) as well as the non-linear relationship between form and function may prevent the detection of direct relationships between morphological rates and rates of evolution in function. Also, the uneven distribution of the relative importance of different functions in phenotypic space means that some functional combinations occupy a restricted area of the morpho-functional adaptive landscape and, hence, they may be less likely to evolve rather than being suboptimal.

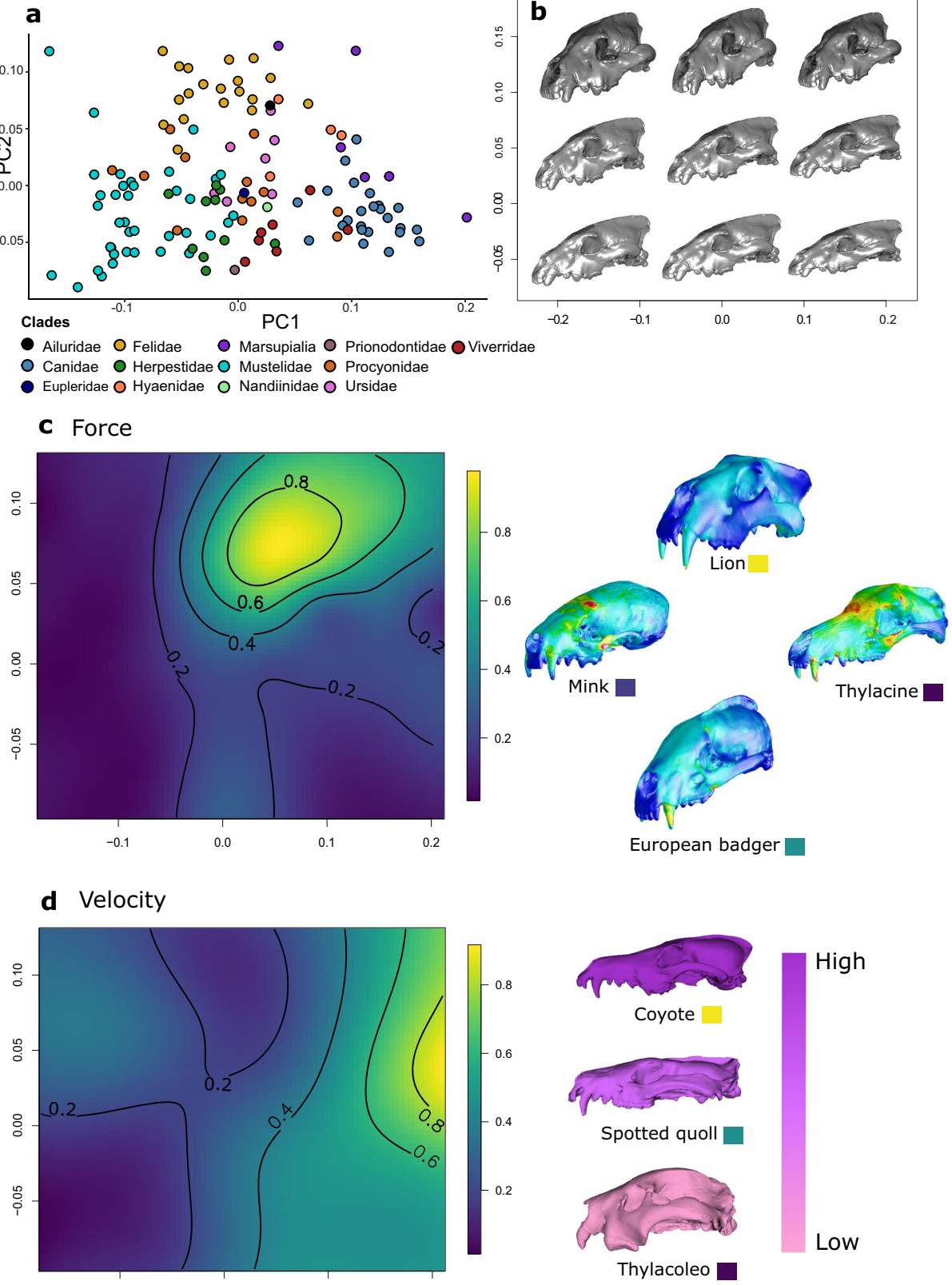

## Results

### Theoretical morpho-functional analysis

We used landmark-based 3D geometric morphometrics to character-ise the shape of 132 carnivore crania. We first performed a principal component analysis (PCA) of the empirical dataset, where the first two axes capture 54.15% of total shape variation, consistently with previous analyses of carnivore skull variation[44,52] (Fig. 1a). We also tested for allometry in carnivore cranial shape while accounting for shared ancestry using phylogenetic generalised least squares (PGLS). We found a weak but significant effect of evolutionary allometry on cranial shape ($r^2 = 0.11$, $P < 0.001$). However, considering that comparable results were obtained regardless of whether we used allometry-free

**Fig. 1 | Shape and functional variation among carnivores. a** PC1/PC2 scatterplot of carnivores' cranial shape variation. **b** Pattern of cranial shape variation projected into the theoretical morphospace. Nine out of 64 theoretical shapes are represented. **c** Performance surface estimated for bite force (estimated as the inverse of von Mises stress; warmer colours on the surface identify areas of higher biting performance) and results from finite element analysis applied to skulls of four species with different levels of bite force (warmer colours on the models identify areas of higher stress). The colour of the boxes near the species' common names corresponds to the value of bite force as displayed on the surface. **d** Performance surface estimated for velocity (estimated as the inverse of mechanical advantage; warmer colours identify areas of higher performance). The models provide three examples of cranial shapes associated to different levels of velocity, with darker colours indicating higher velocity. The boxes near the species' common name identify the value of velocity as displayed in the surface on the left. Source Data are available at https://doi.org/10.6084/m9.figshare.23553648 in the "Code and Source Data" folder, included in the Data_Figure_1.rda file.

shape data or uncorrected data, we employed uncorrected data in downstream analyses.

We generated 64 theoretical shapes using the first two principal component (PC) scores from an evenly spaced 8 by 8 grid constructed on the first two principal components by extending by 10% the range occupied by the empirical shapes, consistently with the proposition that the theoretical landscape should extend beyond the range of observed data[53] (Fig. 1b). Shape variation across the theoretical morphospace thus obtained can be described as a change in overall elongation of the skull moving along PC1 (in our analysis, shapes with negative PC1 scores exhibit comparatively shorter snouts). Along PC2 we can describe variation in overall shortening and dorsoventral flattening of the skull (in our analysis, negative PC2 scores correspond to a more laterally compressed skull).

On the theoretical shapes, we estimated two functional traits subjected to trade-offs: bite force (BF, force as the inverse of von Mises stress), and the inverse of mechanical advantage (IMA, velocity). To estimate these traits, we employed finite elements analysis (FEA) after scaling to unit surface area each of the 64 theoretical shapes to remove the effect of size (see "Methods" section). Then, for each trait (Fig. 1c, d), we fitted a functional surface which represents the relationship between the value of that functional trait and the position in the morphospace described by the first two principal components. To determine the best model representing these functional surfaces, separately for each trait we fitted several models (classes: polynomial, thin plate spline or TPS, and kriging; see "Methods" section for more details), performing 10-fold cross-validation, and choosing the model with lowest root mean square error (RMSE). The BF surface is best characterised by a second-degree TPS (RMSE = 0.042), whereas the IMA surface is best characterised by a third-degree TPS (RMSE = 0.018; see Supplementary Table 1 for full results). We further evaluated the prediction performance of the functional surfaces by comparing predicted functional values with 49 FEA models based on real specimens (see "Methods" section). Both predicted and real BF and IMA were highly correlated (BF Pearson $r = 0.71$, $P < 0.001$; IMA Pearson $r = 0.74$, $P < 0.001$).

Then, to provide an effective measure of the trade-off between BF and IMA, we summed the two functional surfaces and determined across the theoretical morphospace a weighting coefficient $w$, which can be interpreted as the relative importance of each of the two functions at a given value of shape (see "Methods" section). Indeed, the weight $w$, estimated through a likelihood function, is the value that maximises the "functional fitness" of a shape relative to the two functional traits[15,16,28]. Based on the position of each of the species in our dataset, we also used the best-fitting models of functional surfaces described above to estimate the value of the functional traits and, from these, we obtained species-specific values of $w$. The distribution of $w$ among species is skewed so that most of the species in this study display a larger relative contribution of velocity (represented by lower values of $w$). Notable exceptions are felids, bone-cracking hyaenids and bamboo-eating ursids, where bite force has a higher relative importance compared to most other placental species, providing the means to kill relatively large prey and/or process hard materials (See Fig. 2 and Table 1). To test whether the trade-off weight $w$ was due to carnivoran size while accounting for shared ancestry, we performed PGLS which failed to reject the null hypothesis of no effect ($r^2 = 0.16$, $P = 0.49$).

## Trade-offs and rates of morphological evolution

For both shape and trade-off weight (species-specific value of $w$) of placental species in our dataset we fitted a series of evolutionary models, including Brownian motion (BM), early burst (EB) and Ornstein-Uhlenbeck (OU). Overall, BM was the best-supported model of evolution for both trade-off and shape, when compared to OU and EB (see "Methods" section and Supplementary Fig. 1). In addition to these tree-wide models, we fitted, in a Bayesian framework, a variable-rates model allowing for rate shifts in lineages, therefore resulting in branch-specific rates of evolution for shape and the trade-off weight $w$ (Fig. 3a, b). Using Bayes factors, we compared these variable-rates models with a single-rate model (equivalent to BM) and found support for the variable-rates model (Bayes factor greater than 10 for both shape and weight $w$, see Supplementary Figs. 2 and 3).

Morphological and trade-off rates of evolution were largely decoupled as their branch rates were uncorrelated (Pearson $r = 0.014$, $P = 0.82$; Spearman $rho = 0.004$, $P = 0.93$). The same holds for repeating the analysis using tip rates only (Pearson $r = 0.011$, $P = 0.89$; Spearman $rho = 0.002$, $P = 0.97$), and also after accounting for phylogenetic non-independence in tip rates by using phylogenetic independent contrasts (PICs) (Pearson $r = 0.004$, $P = 0.96$; Spearman $rho = 0.09$, $P = 0.27$). Overall, cranial shape evolutionary rates appear more variable within clades than between clades, with episodic accelerations (Fig. 3a, c) as corroborated by a non-significant Mantel test (Z-statistic = 1.99; $P = 0.865$) of the association between the matrix of absolute differences in tip rates and the phylogenetic distance matrix. On the contrary, rates of change in trade-off weight $w$ were more variable between clades than within clades, resulting in more distantly related tips having more distinct rates (Fig. 3b, d; Mantel test: Z-statistic = 2163; $P = 0.007$).

We did not detect any association between shape rates (at the tips) and the value of the trade-off weight $w$. This was tested by fitting three different PGLS models using shape tip rates as the dependent variable and trade-off weight as the predictor. The models were then compared using Akaike's Information Criterion, corrected for small sample size[54] (AICc), which showed how more complex models of the relationship were not better models than an intercept-only model (2nd-degree polynomial AICc = 338.79; linear AICc = 337.9; intercept-only AICc = 337.75). In addition to the lack of association, this result suggests that species showing extreme trade-off weight $w$ values do not show different morphological rates of evolution compared to species showing intermediate ones, a pattern which would have been supported if the 2nd-degree polynomial model were a better fit (Fig. 4). The clustering of $w$ around certain values (Fig. 4) does not allow us to draw conclusions about the – purely hypothetical – case where the distribution of $w$ would not show marked discontinuities.

## Relationship between trade-offs and disparity

To understand the relationship between the force-velocity trade-off and morphological disparity we employed a sliding window approach. First, we defined consecutive, overlapping intervals of weight $w$. Then, we divided the theoretical morphospace described above (first two principal components of shape) using a 40 by 40 grid, with each resulting cell on the functional landscape characterised by a specific value of weight $w$ (Fig. 5a). For each interval (window) of $w$, we calculated the number of cells in the morphospace with values of $w$ in the

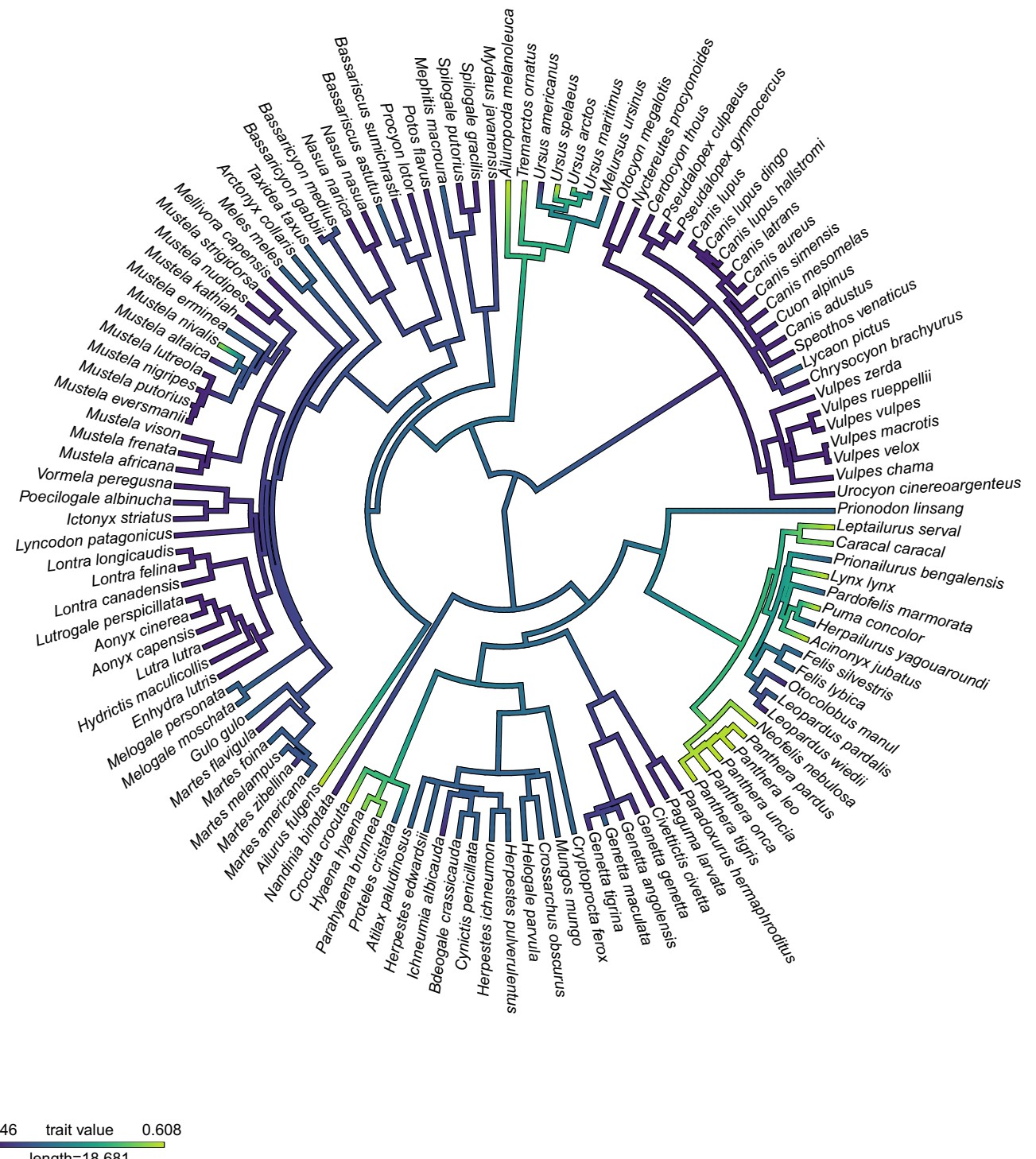

**Fig. 2 | Mapping of trade-off weight *w* on the carnivores' phylogeny.** Warmer colours indicate higher trade-off weight *w* values and cooler colours indicate lower trade-off weight *w* values. Source Data are available at https://doi.org/10.6084/m9.figshare.23553648 in the "Code and Source Data" folder, included in the Data_Figure_2.rda file.

interval. We call this quantity – which represents the "volume" of theoretical morphological space with a given interval of *w* – "weight *w* volume". For each of the intervals of *w*, we also estimated actual morphological disparity (measured using multivariate variance) computed as the disparity of real species in our sample. We then explored whether the pattern of change in weight *w* volume at changing intervals of *w* was similar to the pattern of change in actual disparity. The distributions of trade-off weight *w* volume and disparity showed an overlapping pattern, with three identifiable peaks corresponding to weight *w* values of 0.35, 0.41 and 0.61 (Fig. 5b). Morphological

disparity and weight *w* volume were positively correlated (Pearson $r = 0.51$, $P < 0.001$; Spearman $rho = 0.41$, $P < 0.001$; $xi = 0.42$, $P < 0.001$), meaning that for a given value/interval of *w* (relative importance of one function over the other) the morphospace occupied by actual species was proportional to the theoretically available morphospace characterised by the same range of trade-off weight *w*.

**Morpho-functional prediction for marsupials**

To determine whether marsupials and placentals have similar morpho-functional variation, we used partial least squares (PLS, see "Methods"

**Table 1 | Summary table of nine representative species used to discuss patterns observed in the data**

| Species | Common name | Broad diet | PC1 position | PC2 position | Weight w value |
|---|---|---|---|---|---|
| *Canis latrans* | Coyote | Small prey - hypercarnivore | High values | Low values | 0.346 |
| *Mustela frenata* | Long-tailed weasel | Small prey - hypercarnivore | Low values | Low values | 0.347 |
| *Dasyurus maculatus* | Spotted quoll | Small prey - hypercarnivore | High values | Close to average | 0.347 |
| *Cryptoprocta ferox* | Fossa | Medium prey - hypercarnivore | Close to average | Close to average | 0.418 |
| *Ursus maritimus* | Polar bear | Medium prey - hypercarnivore | Close to average | Low values | 0.418 |
| *Crocuta crocuta* | Spotted hyaena | Bone cracking | Close to average | High values | 0.607 |
| *Panthera tigris* | Tiger | Large prey - hypercarnivore | Close to average | High values | 0.608 |
| *Ailuropoda melanoleuca* | Giant panda | Bamboo | Close to average | High values | 0.608 |
| *Thylacoleo carnifex* | Thylacoleo | Large prey - hypercarnivore | Close to average | High values | 0.607 |

Broad diet categories were assigned following ref. 43. PC1 position and PC2 position refer to position in the morphometric space. Weight w value refers to the trade-ff weight.

section) to model the covariation of shape and performance (i.e., the value of BF and IMA). We first fitted a PLS model on placentals only. Then we computed the expected shapes from this model for both placentals (on which the model had been fit) and marsupials. Finally, we obtained residuals (difference between actual and predicted shape) for both placentals and marsupials. As we used placentals to fit the model, we expect this model to overfit placentals but not marsupials. That is, we expect that residuals for data on which the model has been fitted (placentals, in our case) will be smaller (i.e., closer to the prediction) than the residuals for new data (marsupials). This will be more pronounced if the relationship between form and function differs between carnivorous marsupials and carnivorans (see Methods). Instead, we found no evidence that the association between form and function in marsupials deviates from that detected for placentals. Indeed, when projecting all observations on the space described by the first pair of PLS axes obtained on placentals only, the scores for marsupials are fully inside the "cloud" formed by the scores for placentals (Fig. 6a). Furthermore, the deviations of marsupials from overall patterns of morpho-functional covariation were not larger than those measured for placentals (Fig. 6b).

## Discussion

Trade-offs are generally thought to bias morphological evolution as, by definition, competing functions cannot be simultaneously optimised in anatomical structures. In particular, the bite force-velocity trade-off can have far-reaching evolutionary consequences by determining the organisms' capacity to invade new ecological niches[13,35]. However, we did not find a significant association between rates of evolution in trade-off and morphology (Fig. 3a, b). This might be a consequence of many-to-one mapping and of a non-linear relationship between form and function[14,55–58]. Many-to-one mapping potentially provides multiple pathways for morphological adaptations to achieve a similar functional output. This is well exemplified in our data where similar levels of optimisation of either force or velocity were achieved by taxa showing highly distinct morphologies (e.g., canids and mustelids with similar levels of optimisation for velocity), ecologies (e.g. bone-cracking hyaenids, bamboo-eating ursids and hypercarnivorous felids for force), and/or distinct developmental strategies such as marsupials, which showed trade-off values similar to those of placentals (Figs. 1b and 7 and Table 1). We note that previous studies have demonstrated that mammalian carnivores that commonly take relatively large, less agile prey (e.g., most felids) and/or consume resistant foods, tend to exhibit higher bite forces for their size (hyaenids, bamboo-eating ursids)[43,46]. Most canids and mustelids concentrate on smaller, more agile prey, which likely selects for faster, but less powerful biting. It is notable that a few social canid species do regularly take relatively large prey, but these taxa do exhibit relatively high bite forces compared to other family members of the family[43].

Also, a non-linear relationship between form and function may result in non-univocal changes in shape and performance per unit time[59]. In other words, multiple trait combinations may evolve at very different paces producing similar functional outcomes or else small morphological changes may lead to substantial changes in function[44,60]. Other factors are possibly contributing to the decoupling of rates we observe. Considering the poly-functional nature of the mammalian skull, its morphological variation may be partly driven by environmental (temperature and precipitation) and/or behavioural (hunting strategy, male-male competition) factors[60]. This may, in turn, obscure a hypothetical coupling of rates of morpho-functional evolution related to biting force and velocity.

We did not detect a pattern of different rates of morphological evolution between functionally extreme and less extreme morphotypes (Fig. 4). This finding contrasts the traditional consensus which postulates that functionally extreme morphologies are linked to bursts of phenotypic evolution[19,23,61]. We rather suggest that the relationship between trade-offs and morphological rates of evolution might not be as straightforward as previously thought.

Despite the rates of evolution in trade-off and morphology being largely decoupled, we detected a strong association between trade-off and morphological disparity, although not in the sense that higher levels of relative importance of one of the functions corresponded to higher disparity. Rather, the volume of the theoretical morphospace where force is more important (high weight $w$ values) was dramatically reduced compared to space where velocity has a relatively higher contribution (low weight $w$ values; Fig. 5a). Disparity among species included in our study followed a very similar trend as it peaked at relatively low trade-off values and decreased at higher trade-off values (Fig. 5b). This asymmetric distribution of phenotypic diversity is of particular interest as it offers the chance to understand how species morphological variation can be related to traits subject to trade-offs.

In short, we found that there are often multiple possible shapes available for a specific low-weight $w$ value. That is, selection can generate shapes optimised for bite velocity with many more different trait combinations whereas the same does not apply to high weight $w$ values, where a more limited pool of possible shapes is available to be optimised for bite force.

This pattern may be explained by the presence of constraints inherent to the mammalian skull such as the craniofacial evolutionary allometry (CREA) trend for larger taxa to have proportionally longer faces[62,63], if this allometric pattern is driven by evolution along lines of least resistance[7] as previously postulated. Indeed, one obvious evolutionary pathway to increase bite force is to increase body size[64], but CREA may prevent many lineages from entering areas of the morphospace where force is favoured, as having a relatively longer face tends to increase the out-lever arm, reducing the force output[43,46–48]. In general, skulls adapted to generate high forces adopted similar

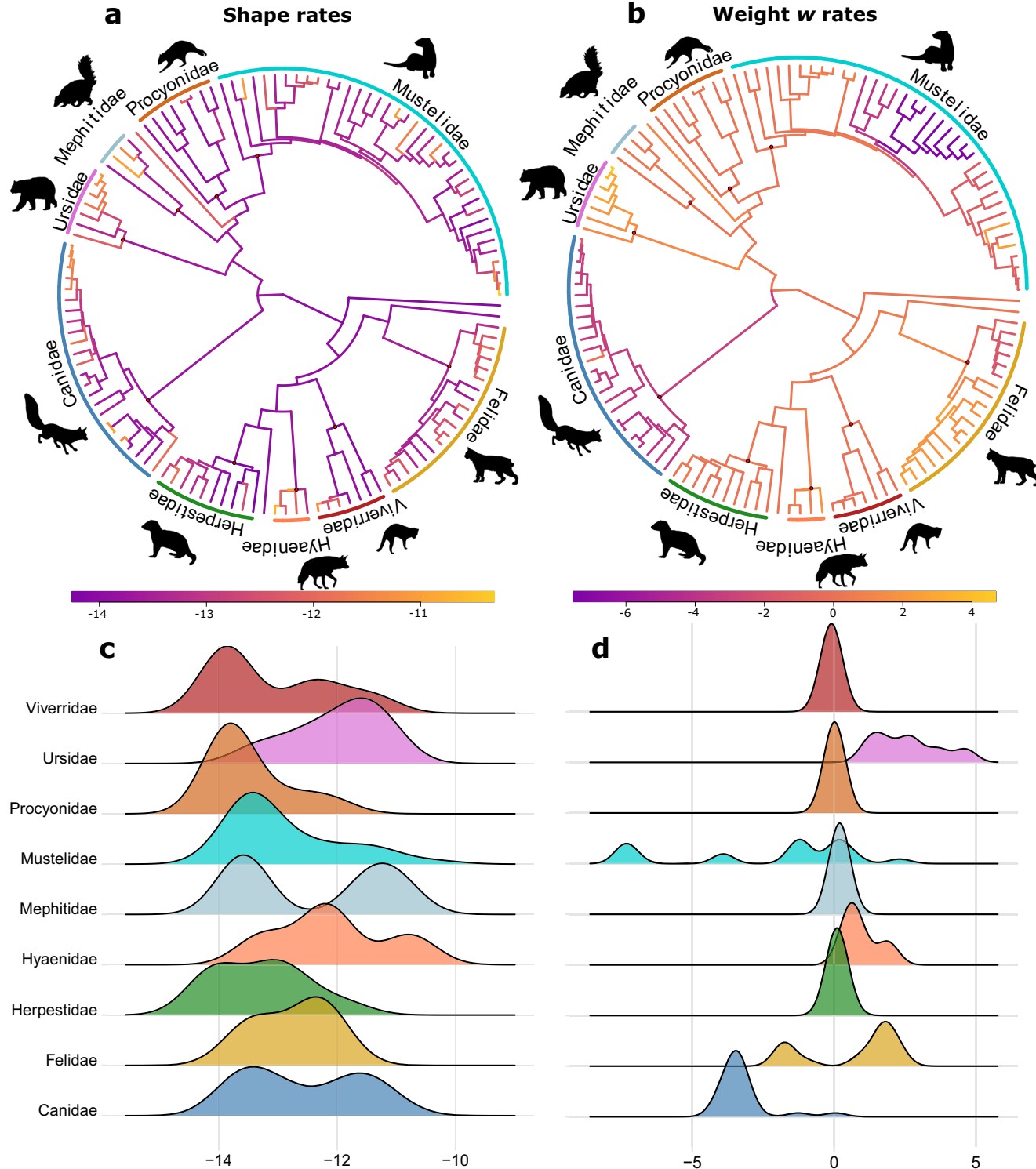

**Fig. 3 | Distribution of cranial shape and trade-off weight w rates of evolution.** **a** Distribution of rates of shape evolution on the placental carnivore phylogeny. Darker colours indicate slower rates and lighter colours indicate higher rates. **b** Distribution of trade-off weight $w$ rates of evolution on the placental carnivore phylogeny. **c** Ridgeline plot displaying per-clade shape rates of evolution. **d** Ridgeline plot of per-clade trade-off weight $w$ rates of evolution. Source Data are available at https://doi.org/10.6084/m9.figshare.23553648 in the "Code and Source Data" folder, included in the Data_Figure_3.rda file.

strategies to overcome the loss of leverage from craniofacial allometry, such as increasing the length of the mandibular coronoid process, increasing muscle physiological cross-sectional area and/or having more robust skulls with thicker bone[50,65–67]. However, these changes in both the external and internal geometry of craniomandibular morphology may come at a cost, further thinning the

range of available solutions to generate these forces. For instance, increasing the length of the coronoid process would reduce the gape, and as a consequence, the capacity to kill relatively large, metabolically valuable, prey (as in the case of large felids[48,68]). Increasing masticatory muscle mass may generate a spatial conflict with brain expansion during growth, as muscle enlargement would limit the effect of brain

expansion on the surrounding cranial bones[69]. Indeed, it has been shown that carnivores with larger temporal muscles have proportionally smaller braincases[70]. Nonetheless, muscles are expensive tissues to maintain, and even more when enlarged considering that muscle force scales to a two-third power rule[17]. Similarly, incorporating

thicker bone could represent a physiological barrier as bone is a metabolically expensive tissue[71]. Here, the relatively large number of these changes, which are also spread across several regions of the cranium, may further decrease the likelihood of their appearance. This reasoning extends also to marsupials, as the species where force is

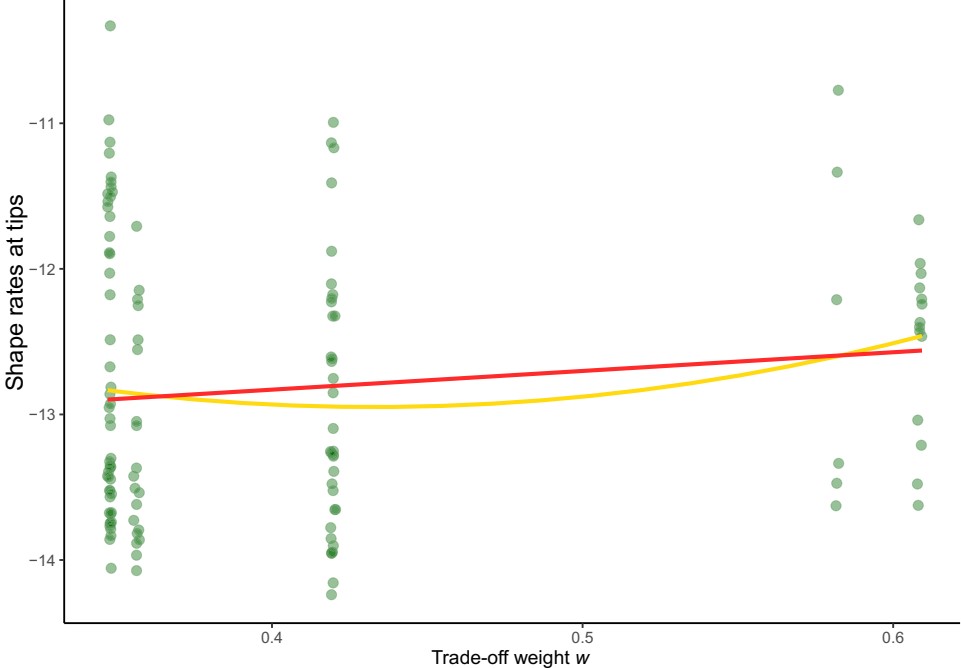

**Fig. 4 | Relationship between cranial shape tip rates and trade-off weight *w*.** The circles represent the values of trade-off weight *w* and cranial shape rate for each species. The two lines represent predictions from regressions of cranial shape rates at tips (dependent) on trade-off weight *w* (independent). The red line is obtained using linear regression, the ochre line using a 2nd-degree polynomial regression. Notice that neither of these models compares favourably to an intercept-only model. Source Data are available at https://doi.org/10.6084/m9.figshare.23553648 in the "Code and Source Data" folder, included in the Data_Figure_4.rda file.

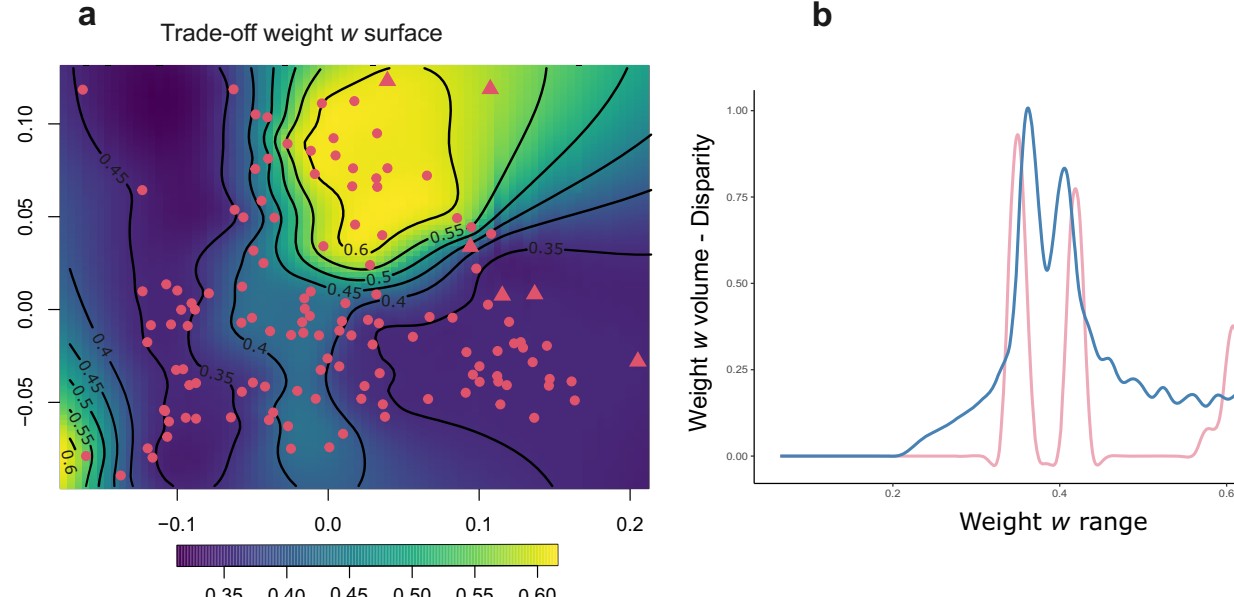

**Fig. 5 | Relationship between morphological disparity and trade-off weight w.** **a** Trade-off weight *w* surface. Warmer colours indicate higher trade-off weight *w* values and cooler colours indicate lower trade-off weight w values. The shapes indicate the position of the real species on the surface. Circles represent placental carnivores, and triangles represent marsupials. **b** Sliding window-based Distribution of actual and theoretical morphological disparity at varying levels of trade-off weight *w* (see "Methods" section for more details). The pink line indicates actual

morphological disparity at a window corresponding to a given weight *w*, and the blue line indicates the weight *w* volume (the theoretically available phenotypic space along the first two principal components at a given weight *w*). To be comparable, the values have been scaled to range between zero and one. Source Data are available at https://doi.org/10.6084/m9.figshare.23553648 in the "Code and Source Data" folder, included in the Data_Figure_5.rda file.

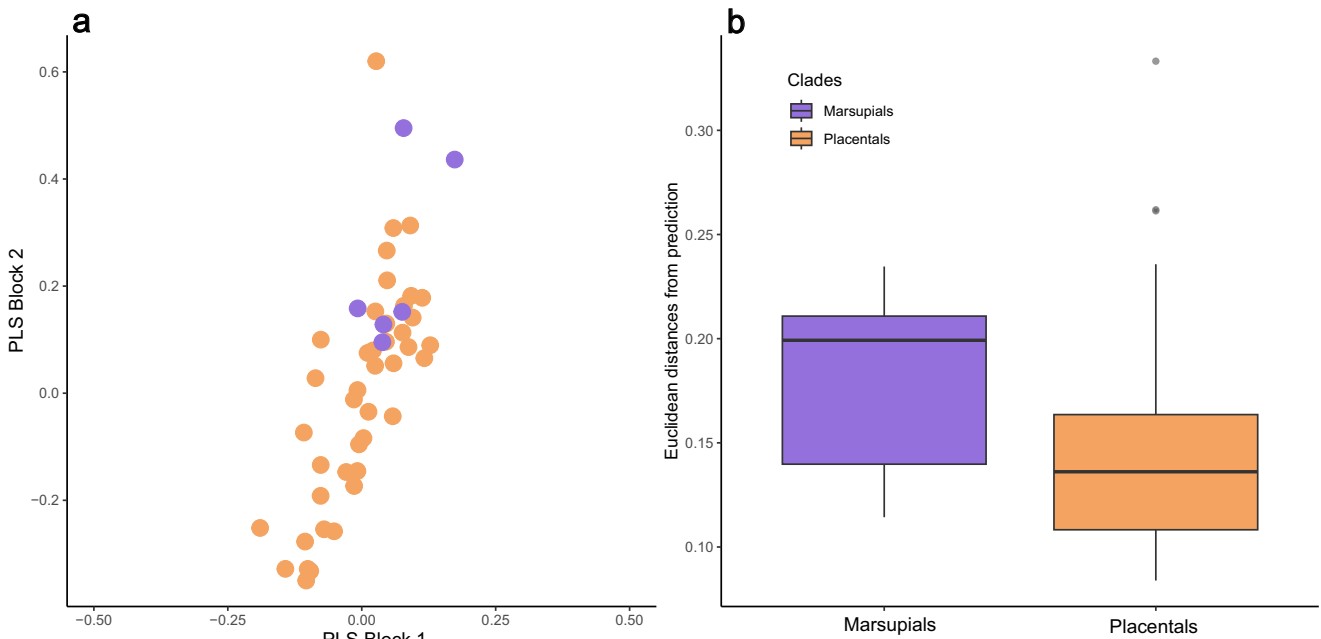

**Fig. 6 | Predictability of marsupial form and function. a** Scores on the first axis of the PLS model fitted on placentals only, orange circles indicate placentals and purple circles indicate marsupials. **b** Euclidean distances between predicted and observed shapes for marsupials and placentals. Source Data are available at https://doi.org/10.6084/m9.figshare.23553648 in the "Code and Source Data" folder, included in the Data_Figure_6.rda file.

more important (the marsupial lion *Thylacoleo carnifex* and the Tasmanian devil *Sarcophilus harrisii* in our sample) are morphologically similar to their placental counterparts (Figs. 1c and 3a). Observing how the same relationship between form and function is found in placentals and marsupials (Fig. 6a, b) – the latter can be considered replicated "experiments" from this point of view – further supports the idea that the force-driven adaptations are constrained biomechanical solutions achieved within a smaller set of available evolutionary pathways as compared to adaptations maximising velocity[66].

In conclusion, evidence for the coupling of form-function rates of evolution remains elusive. However, we propose that this may represent a general feature common to the evolution of complex anatomical structures experiencing many-to-one mapping of form to function. It may also represent a consequence of the highly non-linear relationship between form and function. A similar scenario has been described in teleost fishes' jaws, where a weak association between morphological and functional rates of evolution has been found[9]. Moreover, it is likely that an association between functionally extreme morphologies (i.e., morphologies which markedly optimise one function over another) and increased evolutionary rates may occur in certain contexts (clades, traits) but this association may not be readily generalizable.

Finally, our results argue against the idea that species at extremes of the trade-off should display lower disparity. This is because we observe a highly asymmetric distribution of morphological variability with optimisation of bite velocity at the expense of force associated with large theoretical and realised disparity, and lower disparity at the opposite extreme of the trade-off. Rather, with respect to disparity our results suggest an answer to why we observe relatively few species and morphologies optimising bite force at the expense of velocity. By showing a strong association between the amount of shape variation theoretically and actually found at a given level of the trade-off, the answer is a resounding: because there is a reduced space of theoretical morphologies producing high force and low velocity. We suspect that this type of constraint affecting the amount of phenotypic diversity in nature may be more common than usually thought.

## Methods

### Shape analysis

Mammalian carnivore skull shape variation was quantified by using 35 homologous landmarks (see Supplementary Methods and Supplementary Table 2, and Supplementary Data 1,2 for more details on sampling effort and landmarks digitization procedure). Then, we imported the landmark coordinates into R (v.4.0.1) for further analyses. Using the function procSym from the R package Morpho[72] we performed a generalized Procrustes analysis (GPA) to rotate, translate and scale landmark configurations to unit centroid size, that is the square root of squared differences between landmark coordinates and centroid coordinates[73]. To visualise the variation among species using a two-dimensional projection of the aligned Procrustes coordinates, we used the scores along the first two PCA axes. We classified the 132 species in our sample using similar taxonomic groups as those defined in ref. 52. Then, we tested for the effect of size on shape within a phylogenetic context by performing a multivariate phylogenetic generalised least squares (PGLS) regression of shape coordinates on size (logarithm of centroid size). We found a significant effect of size on shape ($P < 0.001$) with the former explaining 12.25% of variation in the latter. We tested for measurement error by generating three replicates of a subsample of the total dataset (49 species) and performing a repeated measures test[74] which failed to reject the null hypothesis of no effect between the repeated measures ($P = 0.17$; see Supplementary Methods for further details).

### Phylogeny

The phylogeny used in our work is a time-calibrated tree based on a recent, species level, mammalian phylogeny[75]. We downloaded the whole set of 10,000 trees from https://vertlife.org/phylosubsets/ and generated a maximum clade credibility tree using the function MaxCredTree from the R package phangorn[76]. Then, we pruned the maximum clade credibility tree to include only the species represented in our dataset.

### Functional traits

As a proxy for velocity, we used the inverse of mechanical advantage (IMA) as low mechanical advantage is associated with faster force

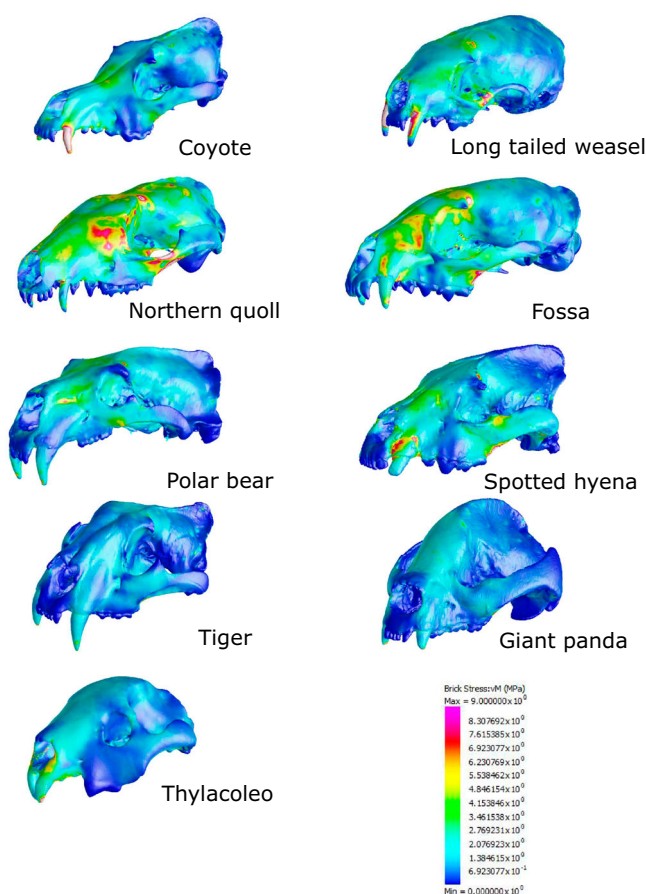

Coyote

Long tailed weasel

Northern quoll

Fossa

Polar bear

Spotted hyena

Tiger

Giant panda

Thylacoleo

**Fig. 7 | Results from finite element analysis.** Nine out of 49 models representative of the carnivores (both placentals and marsupials) von Mises stress variation. The models are not to scale. Von Mises stress follows the same colour scaling for all the models, with warmer colours indicating higher stress, and cooler colours indicating lower stress.

transmission in lever mechanics[34,35,37]. We measured mechanical advantage directly from finite element analysis (FEA, see below) and this quantity was simply defined as the reaction force at the bite point divided by the applied muscle load[37]. As a proxy of bite force (BF) we used the inverse of von Mises stress as shapes capable of sustaining higher loads (e.g. higher bite forces) should display lower von Mises stress[77–79]. Indeed our results confirm this prediction, with taxa showing relatively high bite forces also exhibiting less stress (see Supplementary Methods). However, we also replicated the main analyses presented in the Results section by employing bite force values directly taken from FEA simulations. Results were consistent with those generated using the inverse of von Mises stress and are fully displayed in the Supplementary Methods (see Supplementary Figs. 4–6).

Stress values were averaged across the skull and, to avoid artificially high-stress values at the boundaries and loading points, we removed the top 2% von Mises stress values prior to the computation of the average[37]. To make the two performance metrics comparable we scaled each metric between 0 and 1 following ref. 15.

### Theoretical morphospace

The theoretical morphospace was produced by sampling 64 shapes corresponding to an 8 by 8 grid along the first two PCs (accounting for 55.41% of the total variation). The landmark configurations corresponding to each of the 64 shapes have been reconstructed from scores along the first two PCs using the function restoreShapes from the package Morpho[72]. These landmarks were then used to deform the

mesh corresponding to the species closest to the consensus (in our sample corresponding to *Crocuta crocuta*) following standard protocols[16,80]. The resulting deformed meshes were cleaned and prepared for the subsequent FEA analysis using the software 3Matic (Materialise).

### Finite elements analysis

We generated finite element models (FEMs) for each of the 64 theoretical shapes. Each of the shape warps was assigned homogeneous material properties to simulate cortical bone (Young's modulus = 11,000 MPa, Poisson's ratio = 0.41). Then, we generated volume meshes composed of a similar number of tetrahedral elements (ranging from 1.5 to 1.9 million elements) to facilitate comparison. A surface STL of the mandible (warped in the same space of the cranium, see above) was then imported into the same file as the cranium volume mesh. Furthermore, we checked for potential large departures from mesh homogeneity by computing the PEofAM (Percentage Error of the Arithmetic Mean) and PEofM (Percentage Error of the Median) percentages for our 49 real models and for the 64 theoretical geometries, as proposed in[81] (results are shown in Supplementary Information). Two muscle subgroups representing major jaw-adducting muscle divisions were modelled in each FEM: the temporalis and masseter muscle groups. Specific areas of the mesh were then highlighted to indicate the approximate origin and insertion points for the different muscle subgroups. We did not model the pterygoid muscle as its origin and insertion areas were not obvious in the theoretical models, potentially leading to incorrect estimates.

Following previous studies[82–85], these selected areas were then covered with a network of beam elements to minimise artificial stress singularities and to achieve a more uniform distribution of von Mises stress. These beam elements were assigned a modulus of 200,000 and a Poisson's coefficient of 0.25.

Muscles in the FEMs were modelled as truss elements. A truss element can transmit only axial forces and has three translational degrees of freedom at each node. In each model, a total of 50 truss elements (15 for the temporalis musculature and 10 for the masseter musculature on either side of the skull) were distributed based on their origin and insertion areas. The 3D orientation of muscle trusses was made as symmetrical as possible on both sides of the skull. The surface model of the mandible was used as a tool to guide the connection of muscle beams, with the mandible end of the beams fixed in translation (but not rotation). The width of the muscle beams was standardised between species to remove the body mass effect. For each of the FEMs, intrinsic loadings representing bilateral biting at the canines were applied with the intention of representing behaviours associated with predation and biting. We performed our experiments by simulating carnivores biting at a 35° gape angle, we set this angle equal for each simulation to obtain comparable results, this also applies to the theoretical models (see Supplementary Data 3). The effects of head-neck movement were removed by creating a rigid link in the occipital condyle region. This rigid link was then split into two halves. The centre node was fixed in all six degrees of freedom and acted as an anchor point for the cranium. As size profoundly impacts the biomechanical performance of organisms, we scaled each model to the same surface area following the protocol from ref. 86. Although it would be ideal to include the complex distributions of cortical thickness into FEA models, it has also been demonstrated that meshes with homogeneous internal geometry can accurately predict the biomechanical behaviour of shape[24].

To evaluate the performance of the theoretical models to predict real specimens' performance when mapped onto the theoretical functional landscape (see below), we also generated finite element models (FEMs) for 49 species for which high-resolution CT scans were available (see Supplementary Data 2). The CT cranial data was imported into the segmentation software Mimics (v.21.0) under the DICOM format. The generated volume meshes comprised of tetrahedral elements similar to theoretical models. The volume meshes were then assigned 8

different material properties, to represent the material properties of mammalian bone[87,88] (see Supplementary Table 3). The real specimens were restrained and loaded in the same way as the theoretical ones, and the models were exported and solved into Strand7 (v.3.0.0). Supplementary Figs. 7–55 display the 49 solved models.

### Partial least squares (PLS) analysis of the association between form and function

An important question of this study is whether functional trade-offs may affect shape variation differently between placentals and marsupials. To address this question in an exploratory framework (which is more appropriate due to the relatively small number of marsupial species in our sample) we used as a reference the relationship between functional variables and shape observed in placentals. Because it has been argued that skull shape in marsupials has been constrained by the early postnatal functional demand, it is expected that we should observe a departure from the model of functional relationship observed in the placentals. To test this, we fitted using only placentals a PLS model of the covariation between skull shape (first block) and the two functional metrics scaled to unit variance (second block). Partial least squares analysis[89] seeks the pairs of axes (one for each block) which maximise covariation between blocks (in our case, each pair of axes will describe how shape and function co-vary). In our case, we restricted this analysis to the first pair of PLS axes. Following ref. [90], in addition to obtaining scores along each of the axes on the first pair, major axis regression was used to obtain predictions of the model. In our case, each species (either from placentals or marsupials) can be represented by a score on this major axis (which can be intuitively imagined as an axis passing through the cloud of placental points when plotting scores along the first PLS axes for shape and functional variables). Scores on this major axis can then be back-transformed into shapes and represent the shapes predicted by the partial least squares model. Finally, one can then compute the deviation (as Procrustes distance) between the actual shape of a given species and the shape predicted by the partial least squares model. By obtaining predicted shapes and deviations from the true shapes for both placentals and marsupials but using a model fitted on placentals only we were then able to explore whether the Procrustes distances between predicted and observed shapes for marsupials were on average larger than the distances in placentals. We performed analyses in this section using functions in the R package GeometricMorphometricsMix (https://github.com/fruciano/GeometricMorphometricsMix).

### Functional landscape

To build the functional landscapes based on the 64 theoretical shapes, we first used 10-fold cross-validation to test the fit of different interpolation strategies. We compared (1) polynomial surface fitting up to the fifth degree; (2) Thin plate spline (TPS) surface fitting up to the fourth degree, using the Tps function from the R package fields[91]; (3) Kriging surface fitting, using the autoKrige.cv function from the R package automap[92]. We repeated the cross-validation process for both the IMA and BF metrics, model performance was assessed using the root mean square error (RMSE, see Supplementary Table 3). For MA the best (lowest RMSE) model was a third-degree TPS (RMSE = 0.018), whereas for BF it was a second-degree TPS (RMSE = 0.042). Then, we fitted the two performance surfaces using the chosen TPS function using the first two PC scores as $x$ and $y$ axes and IMA and BF (respectively) as the $z$-axis.

The combined performance landscape can be estimated as the summed contribution of each performance surface following the equation in[15]:

$$\ln \bar{W} = w_1 F_1 + w_2 F_2 + w_n F_n \ldots \tag{1}$$

Where $w_n$ is the weighting coefficient representing the relative contribution of the functional surfaces $F_n$[15,16]. The weighting factor $w$ is a measure of the trade-off between the performance surfaces. In the case of two functions, $w$ can be estimated using likelihood by finding the values of $w$ that maximise the height (position on the $z$-axis) of a specimen on the combined performance landscape $W$, in other words, $w$ defines the relative balance between the two functions at a given position on the combined landscape. The weight $w$ can be computed following the equation in ref. [15]:

$$\ln \frac{w_1 F_1[\bar{z}] + (1 - w_1) F_2[\bar{z}]}{\text{Max}\left[ w_1 F_1 + (1 - w_1) F_2 \right]} \tag{2}$$

Considering that $w_1$ ranges between 0 and 1, and that $w_2 = 1 - w_1$, the weights $w_i$ are the combination of function $F_i$ that best describes the position $(x,y)$ of a given specimen (or shape) on the combined functional landscape.

Furthermore, we assessed the relationship between weight $w$ (as dependent variable) and size (as independent variable) by applying a robust version of PGLS implemented in the R package ROBRT. We used the MM robust estimator which is both robust and resistant to outliers without finding significant results ($P = 0.49$; multiple $r^2 = 0.16$).

### Evolutionary modelling

We assessed the mode of evolution of placentals carnivores cranial shape by fitting three different evolutionary models using the R package mvMORPH[93] using the full set of Procrustes coordinates. We fitted multivariate Brownian motion (BM), Ornstein-Uhlenbeck (OU) and early burst (EB) models of evolution with the function mvgls, which uses a penalised likelihood framework to overcome the limitations of rank-deficient multivariate data such as Procrustes coordinates[94]. We calculated the relative support for each model by means of the extended information criterion[94,95,96] (EIC) which uses semi-parametric bootstrap to estimate the bias (see Supplementary Fig. 1). We assessed the weight $w$ mode of evolution fitting BM, OU and EB evolutionary models using the function fitContinuous from the R package geiger[97], suited to fit likelihood models for continuous univariate traits (see Supplementary Table 4).

### Evolutionary rates

We quantified evolutionary rates for cranial morphology and for the weight $w$ across carnivores using two distinct variable-rates approaches allowing for branch-specific rates computation, implemented in the program BayesTraitsV4.0[98] and in the R package RRphylo[99] (see Supplementary Methods), respectively. Tip rates estimated by BayesTraits and RRphylo were highly correlated for each of the considered "traits" (see Supplementary Table 5), hence, considering the replicability of the results generated by the two approaches and for the sake of readability, only the results of BayesTraits will be presented and discussed.

BayesTraits employs a reversible jump MCMC chain to estimate the probability of rate shifts in phenotypic trait data across a phylogeny. The method uses Bayesian inference to fit evolutionary models of evolution in phenotypic traits given a phylogenetic hypothesis and allows comparing models using Bayes factors. We compared two classes of models of trait evolution (i) all lineages evolve under a Brownian motion model and share a single rate of evolution (single rate Brownian motion); (ii) all lineages evolve under a Brownian motion model, but the rate of evolution varies across lineages and taxa (variable-rates Brownian motion). In the case of shape, to reduce data dimensionality (and BayesTraits runtime) and to obtain evolutionarily uncorrelated variables starting from multivariate shape data, we first run a phylogenetic principal component analysis[100] and used the scores along the first twenty phylogenetic principal components, accounting for 90.51% of the total variation. To account for phylogenetic uncertainty, we repeated the analysis using a distribution of 1000 different phylogenetic hypotheses (randomly sampled from the 10,000 trees derived from ref. [75] as described above) as input. We

used 500,000,000 iterations and a burn-in of 50,000,000 iterations and the chain was sampled every 100,000 iterations using a stepping-stone sampler. Each analysis was carried out twice and convergence between chains was assessed using Gelman and Geweke diagnostics as implemented in the R package coda[101]. We compared the likelihood of each model using Bayes factors computed using the R package BTprocessR (https://github.com/hferg/BtprocessR). Variable-rate Brownian motion is strongly favoured over single-rate models (Bayes factor greater than 10; see Supplementary Fig. 2). We repeated the same procedure on the weight $w$ (see Supplementary Fig. 3).

Finally, we tested whether cranial shape and weight $w$ evolutionary rates of tree branches were associated using linear models, and whether rates at tips were associated by applying a robust version of phylogenetic generalized least squares regression (PGLS) implemented in the R package *ROBRT*.

### Analysis of morphological disparity

To understand the relationship between cranial disparity and the force-velocity trade-off we employed a sliding window approach. First, we defined consecutive, overlapping windows over the weight $w$, generating a sequence of 3126 windows with a step size of 0.0002 and a window size of 0.011. Then, we generated a 40 by 40-point grid to increase resolution following the previously described procedure. As each point (or cell) on the functional landscape is defined by a specific value of weight $w$, we calculated the number of points with values of $w$ in each window. This provides an estimate of the volume at weight $w$, that is how variable are the theoretically possible shapes in the morphospace with that range of $w$. Finally, we identified the intervals populated by at least two real species to measure actual morphological disparity in that interval. However, similar weight w values can occur in distant areas of the landscape and estimating disparity within intervals populated by species positioned in very distant areas of the landscape would greatly overestimate the disparity for that interval. To avoid this issue, we used – for each interval – a clustering approach. We first used the clustering algorithm implemented in the R package mclust[102] to cluster species in a given interval of $w$ based on their position in the theoretical morphospace, we then estimated the disparity separately for each cluster of species and, finally, we summed the disparity across clusters. We used as estimator of morphological disparity multivariate variance (sum of univariate variances) and we computed it across all shape variables (rather than along the first two principal components only). Finally, we explored the relationship between weight $w$, volume of weight $w$, and actual disparity by plotting them (see Fig. 5a, b) and by computing correlations between weight $w$ volume and disparity by means of Pearson $r$, Spearman *rho*, and *Xi* coefficients[103].

### Reporting summary

Further information on research design is available in the Nature Portfolio Reporting Summary linked to this article.

## Data availability

All data, including biomechanical simulations, required to replicate this study are available at https://doi.org/10.6084/m9.figshare.23553648. Shape data performance data are available in the DR_shape.rda and in the DR_w.rda file. Source data are provided within the codes included in the file Code_Rscript.r. Within the script, there are instructions to load separate.rda file that allow to reproduce Figs. 1–6 and Supplementary Figs. 1, 4–6. Specimen accession codes are available in Supplementary Data 1 and 2.

## Code availability

The code required to replicate this study is included in the file Code_Rscript.r which is available at https://doi.org/10.6084/m9.figshare.23553648.

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

## Acknowledgements

We are grateful to Pasquale Raia, Silvia Castiglione and Carlo Meloro for their useful comments during the manuscript preparation. We are grateful to three anonymous referees for their thoughtful comments. SW received funding from the Australian Research Council Discovery Projects DP140102659. G.S. was partially supported by the Ministry of University and Research of Italy (MUR), project FOE 2020-Capitale naturale e risorse per il futuro dell'Italia.

## Author contributions

G.S., S.W. and C.F. conceived the study. G.S. and C.F. analysed the data. G.S., G.C. and M.A. collected the specimens and performed the biomechanical simulations. G.S. digitised the landmarks. G.S., C.F. and S.W. wrote the manuscript.

## Competing interests

The authors declare no competing interests.
