## [Peer Review File · Nature Communications]

Unexpectedly uneven distribution of functional trade-offs explains cranial morphological diversity in carnivoresReviewers' Comments:

Reviewer #1:

Remarks to the Author:

This manuscript describes the effect of the trade-off between force and velocity in biting on the evolution of cranial morphology in mammalian carnivores, including carnivorans and carnivorous marsupials. The authors conducted this study using a combination of elaborate methods. They found no significant association between the evolutionary rates of cranial morphology and the relative importance of velocity and force in the trade-off. They also found that cranial shape exhibited greater variation when optimizing for velocity at the expense of force.

I did not find any major issues in this manuscript. I find the findings of this study interesting. The evolution of skull morphology has captured the attention of numerous researchers. Furthermore, constraints on the evolution of morphological traits due to trade-offs are a major topic in the field of evolutionary biology, and this study could have important implications for this topic.

In this paper, I would like the authors to discuss the bite forces estimated in this study with reference to those from other studies. Previous studies (e.g., Christiansen & Adolfssen 2005; Wroe et al. 2005; Christiansen & Wroe 2007) have compared the estimated bite forces among carnivores. Moreover, variation in the evolutionary rate of bite force has been previously investigated (Sakamoto et al. 2019). It may be beneficial to compare the findings and methods of this study with those of these studies.

References:

- Christiansen, P., Adolfssen, J. S. (2005). Bite forces, canine strength and skull allometry in carnivores (Mammalia, Carnivora). *J. Zool.* 266:133–151.
- Wroe, S., McHenry, C., Thomason, J. (2005). Bite club: comparative bite force in big biting mammals and the prediction of predatory behaviour in fossil taxa. *Proc. R. Soc. B.* 272:619–625
- Christiansen, P., Wroe, S. (2007). Bite forces and evolutionary adaptations to feeding ecology in carnivores. *Ecology* 88: 347-358.
- Sakamoto M, Ruta M, Venditti C. (2019). Extreme and rapid bursts of functional adaptations shape bite force in amniotes. *Proc. R. Soc. B* 286: 20181932.

Other comments:

L.29 "macroevolutionary patterns of morphological and ecological variation"

L.548 "evolutionary models of variation in phenotypic traits"

I feel these expressions somewhat incongruous. Morphological and ecological (phenotypic) traits evolve, not variation.

L. 80 "a single functional metric"

Is this a metric of one function or a metric of a combination of two functions?

L. 340 – 365

The bite force depends on a number of features such as leverage, length of the mandibular coronoid process, and muscles physiological cross-sectional area, while the velocity that was estimated as the inverse of mechanical advantage may be determined by only a few elements of cranium. In this case, optimizing for velocity is expected to have no effects on many parts of the cranium, allowing for a variety of cranial shapes. This may provide a simpler explanation for the results than the one argued in this manuscript.

L.395

This sentence appears to be incomplete. Perhaps some phrase should be inserted after the word "in".

L.458, L. 483, and so on.

Would it not be better to insert some words instead of just the reference number?

L.524

A period instead of a comma?

Extended Data Table 1.

Suggest give a reference to "Broad diet". The diet of extinct species should be based on inference.

Reviewer #2:

Remarks to the Author:

In think that this is an interesting study that focuses on an important aspect on how organisms evolve and diversify. The study tackles a key aspect of these processes, trying to explain how the form and function of the mammalian skull interact and ultimately produce the disparity we observe in the different groups today. In particular, the authors focus on the trade-offs occurring in the jaw of carnivore taxa and how it impacts on the rate of evolution observed in different clades. I think that the contribution is important and original and should be of interest to a wide spectrum of researchers. I have some minor concerns that I think the authors should correct or describe for the manuscript to be published in Nature Communications.

There are two specific things that in my opinion should be corrected in the manuscript.

Regarding the FEA and the evaluation of the results, the authors chose to use the average von Mises stress value across the skull after the exclusion of the upper 2% (a procedure common in this kind of analysis and correctly justified by the authors). However, the authors should account for the homogeneity of the mesh regarding element size and its impact on the results. In this sense, the use of non-homogeneous meshes to derive statistics like the average can pose certain difficulties. I urge the authors to analyze their meshes for homogeneity regarding element size and report those results or use the corrected statistics proposed by Marcé-Nogué et al. (2016): <https://palaeo-electronica.org/content/2016/1548-statistical-approach-of-fea>.

Regarding the use of AIC and EIC for the selection of the best fitting evolutionary models, there are some inconsistencies in the text that need to be corrected. In particular, the authors state that the best fitting model in both cases was the BM model, but Figure 1 in the Supp. Material seems to point to an EB model as the one with the lower EIC in the case of shape.

On the other hand, the AIC values reported for the different PGLS models for the evaluation of the relationship between shape rates vs w are extremely close and do not show preference for any model. In this case, I don't think the results support the interpretation of the authors regarding a lack of a relationship between these variables. In fact, in Extended Data Figure 2 it becomes clear that the data do not cover all the spectrum of w values and there is a lot of spread in rate values, which clearly makes any interpretation of the potential relationship a problem, as the AIC clearly shows that a linear model is practically equal to an intercept one for explaining the data. The authors should seriously consider their interpretation and at least provide a better interpretation of the results pointing out its potential weaknesses.

Reviewer #3:

Remarks to the Author:

Dear Authors and Editors

In this study Sansalone and collaborators explore the interplay between morphological evolution and functional trade-off in the musculoskeletal system using the cranium of carnivore mammals as a case

study. Specifically, the authors aim to address two questions: whether there is a correlation between the force-velocity trade-off in the jaw-closing system and the rates of functional and morphological evolution and how trade-offs influence morphological disparity.

The study is based on a large sample (132 species encompassing 12 families of terrestrial carnivores), and combines shape analysis using 3D geometric morphometrics, biomechanical modelling (finite element analysis) and evolutionary modelling. Hence, the scope, sample and approach of the study has the potential to generate significant results for our understanding of mammal evolution. The manuscript is very well-written, and the authors have done a great job of explaining the different steps of their work and the complicated methods they used. However, I think there are serious issues with the biomechanical analyses presented in this work, in terms of the quality of the data and level of support of the conclusions. This is very problematic as the functional variables generated by the biomechanical models are then used in the downstream evolutionary analyses, and therefore impact the conclusions made on the morpho-functional evolution.

This paper explores the trade-off between bite force and jaw-closing speed in the feeding system of carnivore mammals. Bite force and jaw-closing speed are two metrics that can be estimated by measuring the length of lever arms of the muscles (distance between muscle insertion and temporomandibular joint [TMJ]) and the out-lever of the mandible (distance from the TMJ to the bite point). Using 3D surfaces of the cranium and mandible, this can be easily achieved in a software like AVIZO or Blender. Provided there are data on muscle PCSA or forces, these measurements can be used to make static models to calculate moments at the TMJ and reaction force at the bite point – something that can be done in a software like Excel. Hence, I am not sure why the authors used such a complicated approach as finite element analysis (FEA) to calculate bite force and jaw-closing speed in numerous species (L140, L183-186)? From a biomechanical standpoint, the method employed to address the question in this paper is thus not adequate.

This issue is further exacerbated by the way the biomechanical analyses have been performed, which makes me quite sceptical about their biological accuracy. Again, this is a very important problem here as the downstream analyses are based on the functional variables extracted from those biomechanical models.

First, I do not understand why the authors used the inverse of the von Mises stress (L416) as an estimate of bite force while it can be directly computed as the reaction force at the constrained node on the teeth (L415)? Moreover, I think that the authors' argument to justify this approach ("shape capable of sustaining higher loading conditions should display lower von Mises stress" [L416-417]) is also problematic. This assumes that the cranium is, as a whole unit, optimised (or not) to resist feeding loads (stress value are here averaged across the whole skull [L418]), while we know that mammalian cranium experience marked bone strain (and therefore stress) gradients during biting. Some areas of the cranium such as the bow ridge in primates experience very low strain magnitudes during biting compared to others, suggesting that bone mass in those areas is unrelated to the resistance to feeding loads (i.e. they are overdesigned to resist those loads) (e.g. Godinho et al. 2018 *Nature Ecol. Evol.*). Therefore, as the cranium performs many functions, there might not be a single optimality criterion in the design of different regions of the cranium of the same animal, which make the relationship between the loading regime (the combination of forces applied to the skull during biting) and bone distribution in the cranium far more complex than what is presented here (see for discussion Ross et al. 2019 *J. Exp. Biol.*; Hylander & Johnson 1997 *Am. J. Biol. Anthropol.*). Therefore, I do not think that the author employed the correct approach to estimate their functional variables, as what really matters for the trade-off studied here is the musculoskeletal function (how muscle forces are generated and transferred) rather than the structural behaviour (how the skeleton resists those loads) of the system.

Then, only the temporalis and masseter groups are modelled (L438-440), and at no point in the manuscript do the authors explain why the pterygoid muscle group was not included. I am sorry if I

have missed something, but I do not understand the rationale behind this choice, especially as it likely has consequences on the outputs of the models. Also, there is little information as to how the muscle orientations were reconstructed; for instance, does the temporalis muscle wrap onto the cranial vault or was it modelled as following a straight path? The way muscles are reconstructed indeed affects reaction forces (see for instance Groening et al. 2013 J. Royal Soc. Interface).

Moreover, this is also very not clear to me how the authors estimated muscle force in the 49 'real' specimens (L432-469). This is quite problematic as muscle morphology and architecture (muscle volume, fibre lengths, pennation angle) are important here; for instance, increasing muscle size (with fibre length unchanged) or decreasing fibre length (with muscle size unchanged) while maintaining the same lever arms would result in greater bite force. Moreover, there is no information on the gape at which maximal bite force was estimated, while we know that the configuration of the muscles and the length-tension relationship of sarcomeres impact the gape at which maximal bite force is generated. I am therefore quite concerned that we miss quite a lot here about the musculoskeletal function, which eventually skews the conclusions about functional trade-offs between bite force and jaw-closing speed.

For taxa where published data are available, I would have appreciated to see how the relative differences in bite force between taxa predicted with FEA reflects those observed with in vivo bite force measurements. This would have allowed the authors to demonstrate the biological accuracy of their 49 FE models, which were used as a reference to estimate the accuracy of the FEAs made on the functional surfaces (L194-197).

For all the reasons detailed above, I have very little confidence in the biomechanical models presented here, and, as a consequence, in the functional variables underpinning the conclusions on the morpho-functional evolution of the carnivore feeding apparatus. Unfortunately, and despite the interesting question addressed and the scope of the study, I cannot support this article for publication in Nature Communications.

Reviewer #1 (Remarks to the Author):

This manuscript describes the effect of the trade-off between force and velocity in biting on the evolution of cranial morphology in mammalian carnivores, including carnivorans and carnivorous marsupials. The authors conducted this study using a combination of elaborate methods. They found no significant association between the evolutionary rates of cranial morphology and the relative importance of velocity and force in the trade-off. They also found that cranial shape exhibited greater variation when optimizing for velocity at the expense of force.

I did not find any major issues in this manuscript. I find the findings of this study interesting. The evolution of skull morphology has captured the attention of numerous researchers. Furthermore, constraints on the evolution of morphological traits due to trade-offs are a major topic in the field of evolutionary biology, and this study could have important implications for this topic.

Answer: We thank the reviewer for their very positive general assessment of our work and its broad importance.

In this paper, I would like the authors to discuss the bite forces estimated in this study with reference to those from other studies. Previous studies (e.g., Christiansen & Adolfssen 2005; Wroe et al. 2005; Christiansen & Wroe 2007) have compared the estimated bite forces among carnivores. Moreover, variation in the evolutionary rate of bite force has been previously investigated (Sakamoto et al. 2019). It may be beneficial to compare the findings and methods of this study with those of these studies.

References:

Christiansen, P., Adolfssen, J. S. (2005). Bite forces, canine strength and skull allometry in carnivores (Mammalia, Carnivora). *J. Zool.* 266:133–151.

Wroe, S., McHenry, C., Thomason, J. (2005). Bite club: comparative bite force in big biting mammals and the prediction of predatory behaviour in fossil taxa. *Proc. R. Soc. B.* 272:619–625

Christiansen, P., Wroe, S. (2007). Bite forces and evolutionary adaptations to feeding ecology in carnivores. *Ecology* 88: 347-358.

Sakamoto M, Ruta M, Venditti C. (2019). Extreme and rapid bursts of functional adaptations shape bite force in amniotes. *Proc. R. Soc. B* 286: 20181932.

Answer: Thank you for the suggestion. We are aware that estimates can vary across studies and how beneficial it can be to add such checks of consistency with other studies.

We added a table in Supplementary Information (Supplementary Table 2) comparing our bite force estimates with those from other published studies. As the Reviewer will note, our results are fairly consistent with previously published results.

We believe that our quantitative comparisons with earlier estimates of bite force result in further confidence in the robustness of our results.

The paper from Sakamoto et al. (2019) analysed the evolutionary rates of bite forces collected from multiple published investigations (including some of the papers indicated below by the reviewer). This is a very different approach to ours, i.e., measuring the rates of evolution of the parameter w which quantifies trade-off between bite force and velocity. Therefore, a direct comparison of the results would not be appropriate.

Furthermore, the study from Sakamoto et al. (2019) focuses on a larger phylogenetic scales rather than focusing on a specific clade as we did.

Other comments:

L.29 “macroevolutionary patterns of morphological and ecological variation”

L.548 “evolutionary models of variation in phenotypic traits”

I feel these expressions somewhat incongruous. Morphological and ecological (phenotypic) traits evolve, not variation.

Answer: Thank you. We corrected the text following the reviewer’s suggestions.

L. 80 “a single functional metric”

Is this a metric of one function or a metric of a combination of two functions?

Answer: We are referring here to a metric of one function. We edited the text to make it clearer. See lines 96-98. “a single metric is used, with extreme values corresponding to optimizing one of two functions, which constrains along a single axis the range of possible combinations between the two functions”.

L. 340 – 365

The bite force depends on a number of features such as leverage, length of the mandibular coronoid process, and muscles physiological cross-sectional area, while the velocity that was estimated as the inverse of mechanical advantage may be determined by only a few elements of cranium. In this case, optimizing for velocity is expected to have no effects on many parts of the cranium, allowing for a variety of cranial shapes. This may provide a simpler explanation for the results than the one argued in this manuscript.

Answer: We discussed this aspect in text. Basically, in the same lines cited by the reviewer, we explored the possibility that increasing bite force would require modifying multiple traits, such as muscle size and position, thickening the bone and/or broadening the skull. Whereas, increasing velocity may occur, in example, simply by lengthening the rostrum. However, we did not consider these motivation as a whole, as suggested by the reviewer. Hence, to further elucidate this aspect, we added a sentence in the discussion stating: “Nonetheless, the relatively large number of these changes, which are also spread across several regions of the cranium, may further decrease the likelihood of their appearance”. See lines 371-373.

L.395

This sentence appears to be incomplete. Perhaps some phrase should be inserted after the word "in".

Answer: Sorry, we forgot to add a reference after the word “in”, specifically: Law et al. 2022.

L.458, L. 483, and so on.

Would it not be better to insert some words instead of just the reference number?

Answer: We prefer to maintain this format as it makes the text more streamlined.

L.524

A period instead of a comma?

Corrected.

Extended Data Table 1.

Suggest give a reference to “Broad diet”. The diet of extinct species should be based on inference.

We added the reference for the broad dietary categories. That is Wroe, S., McHenry, C., & Thomason, J. (2005). Bite club: comparative bite force in big biting mammals and the prediction of predatory behaviour in fossil taxa. *Proceedings of the Royal Society B: Biological Sciences*, 272(1563), 619-625.

Reviewer #2 (Remarks to the Author):

In think that this is an interesting study that focuses on an important aspect on how organisms evolve and diversify. The study tackles a key aspect of these processes, trying to explain how the form and function of the mammalian skull interact and ultimately produce the disparity we observe in the different groups today. In particular, the authors focus on the trade-offs occurring in the jaw of carnivore taxa and how it impacts on the rate of evolution observed in different clades. I think that the contribution is important and original and should be of interest to a wide spectrum of researchers. I have some minor concerns that I think the authors should correct or describe for the manuscript to be published in Nature Communications.

Again, we thank the reviewer for this positive evaluation of our manuscript in terms of its general interest.

There are two specific things that in my opinion should be corrected in the manuscript.

Regarding the FEA and the evaluation of the results, the authors chose to use the average von Mises stress value across the skull after the exclusion of the upper 2% (a procedure common in this kind of analysis and correctly justified by the authors). However, the authors should account for the homogeneity of the mesh regarding element size and its impact on the results. In this sense, the use of non-homogeneous meshes to derive statistics like the average can pose certain difficulties. I urge the authors to analyze their meshes for homogeneity regarding element size and report those results or use the corrected statistics proposed by Marcé-Nogué et al. (2016): <https://palaeo-electronica.org/content/2016/1548-statistical-approach-of-fea>.

Answer: We thank the reviewer for this suggestion. We accounted for the potential discrepancies in mesh homogeneity by computing the two indices proposed by Marcé-Nogué et al. (2016). Specifically, we computed the *PEofAM* (Percentage Error of the Arithmetic Mean) and *PEofM* (Percentage Error of the Median) percentages for our 49 real models and for the 64 theoretical geometries. Results were that our meshes are sufficiently homogeneous as they ranged between 0.21% and 2.36% (median = 0.83%) for

***PEofAM* and between 1.78% and 5.22% (median = 2.53%) of *PEofM*, suggesting that raw stress data could be safely used for further statistical analyses. We want to stress that these results are not surprising considering that we controlled for the mesh elements size during our procedure of mesh preparation before FEA. This procedure is carried out in 3Matic software ensuring that the different meshes are composed not only by a similar number of elements (as detailed in the text) but also by elements of similar size. This procedure allows for FEA results to be comparable and avoids that peak stress values could be artifactually concentrated on smaller elements.**

For the sake of completeness, we added the results detailed above in Supplementary Information in the section dedicated to FEA analysis.

Regarding the use of AIC and EIC for the selection of the best fitting evolutionary models, there are some inconsistencies in the text that need to be corrected. In particular, the authors state that the best fitting model in both cases was the BM model, but Figure 1 in the Supp. Material seems to point to an EB model as the one with the lower EIC in the case of shape.

Answer: We want to reassure the reviewer regarding our interpretation. The EIC allows for model comparison, and similarly to the most commonly employed AIC the lower the EIC the “better”. However, two important differences are 1. that the bias term used in computing EIC is estimated through bootstrap simulations rather than by using analytical or approximate solutions (detailed in the text Line 688; see also the help for the *mvMORPH* package) and 2. that one cannot use the common “rule of thumb” (used for AIC) of “2 units difference”. Because of these two differences with AIC, the appropriate way to compare EIC across models is to compare distributions of the EIC (as the bias term is estimated through bootstrap, then also EIC will have a distribution obtained through bootstrap). In Extended Data Figure 2 we plot for each model both the point estimate (black dot) and +/- 2SE. The rationale is that if the intervals are overlapping, then they are “indistinguishable” in terms of support (therefore, there is no advantage in choosing a more complex model over a simple model). In the plot, despite the EB point estimate being the lowest compared to those of the BM and OU models, the interval of EIC for the EB model considerably overlaps with the bias of both BM and OU models. Therefore, we can conclude that there is no special reason to choose a more complex model (EB or OU) over a simpler model (BM).

To further clarify this, we have updated the text of the legend.

On the other hand, the AIC values reported for the different PGLS models for the evaluation of the relationship between shape rates vs w are extremely close and do not show preference for any model. In this case, I don't think the results support the interpretation of the authors regarding a lack of a relationship between these variables. In fact, in Extended Data Figure 2 it becomes clear that the data do not cover all the spectrum of w values and there is a lot of spread in rate values, which clearly makes any interpretation of the potential relationship a problem, as the AIC clearly shows that a linear model is practically equal to an intercept one for explaining the data. The authors should seriously consider their interpretation and at least provide a better interpretation of the results pointing out its potential weaknesses.

We entirely agree with this interpretation, but we do not see how it is, in practice, different from ours. As you correctly point out, there is no support for a more complex model (either a 2nd degree polynomial or a linear model with a slope) compared to a simpler intercept-only model. But an intercept-only model means a horizontal line implying that a change in the predictor (w) does not

result on average in a change in the dependent variable (shape rates). This is exactly what we would expect in the case of no relationship between the two, which is how we interpret our results.

With respect to the spread of shape rates, we think that this simply reinforces the idea of a relationship not being present or, at least, not being strong (or not having strong predictive value). Let us assume that having much larger data and, consequently, larger sample sizes in statistical analysis and that this would result in a “significant” relationship between shape rates and w as a consequence of increased statistical power. In this case, the relationship would anyway have limited predictive/practical value as there would be extremely large scatter around the prediction line. In our case, however, we do not even detect a relationship.

With respect to the fact that our observations do not cover every possible value of w , we agree that in principle this may pose inferential problems. However, this is the empirical data we observe. Please note how we decided to fit a second-degree polynomial to reflect the idea that has been proposed that functional extremes will have different rates than functional intermediates. In any event, to address this concern, we added a short sentence at 268-270, but we stress that the case of a distribution of w without discontinuities would be a purely hypothetical. To understand why we think this is a purely hypothetical case, note how to do the same plot with intermediate values of w we would need, in addition to shapes corresponding to intermediate values of w , also their position in an actual empirical phylogeny so that we could estimate shape rates. Considering that our empirical dataset is already quite large, we consider it unlikely that the same analysis run adding the missing species would result in a much better coverage of possible values of w .

Reviewer #3 (Remarks to the Author):

Dear Authors and Editors

In this study Sansalone and collaborators explore the interplay between morphological evolution and functional trade-off in the musculoskeletal system using the cranium of carnivore mammals as a case study. Specifically, the authors aim to address two questions: whether there is a correlation between the force-velocity trade-off in the jaw-closing system and the rates of functional and morphological evolution and how trade-offs influence morphological disparity.

The study is based on a large sample (132 species encompassing 12 families of terrestrial carnivores), and combines shape analysis using 3D geometric morphometrics, biomechanical modelling (finite element analysis) and evolutionary modelling. Hence, the scope, sample and approach of the study has the potential to generate significant results for our understanding of mammal evolution. The manuscript is very well-written, and the authors have done a great job of explaining the different steps of their work and the complicated methods they used. However, I think there are serious issues with the biomechanical analyses presented in this work, in terms of the quality of the data and level of support of the conclusions. This is very problematic as the functional variables generated by the biomechanical models are then used in the downstream evolutionary analyses, and therefore impact the conclusions made on the morpho-functional evolution.

As you will see below, we have made substantial effort to address the reviewer’s concerns. We think that this has allowed us to further ensure that our results and interpretations are.

This paper explores the trade-off between bite force and jaw-closing speed in the feeding system of carnivore mammals. Bite force and jaw-closing speed are two metrics that can be estimated by measuring the length of lever arms of the muscles (distance between muscle insertion and temporomandibular joint [TMJ]) and the out-lever of the mandible (distance from the TMJ to the bite point). Using 3D surfaces of the cranium and mandible, this can be easily achieved in a software like AVIZO or Blender. Provided there are data on muscle PCSA or forces, these measurements can be used to make static models to calculate moments at the TMJ and reaction force at the bite point – something that can be done in a software like Excel. Hence, I am not sure why the authors used such a complicated approach as finite element analysis (FEA) to calculate bite force and jaw-closing speed in numerous species (L140, L183-186)? From a biomechanical standpoint, the method employed to address the question in this paper is thus not adequate.

Answer: The reviewer is certainly right as bite forces can be estimated through 2D methods (see Wroe et al. 2005). However, these methods rely on two major simplifications: 1) they treat the skull as a beam; 2) the placement of each muscle force at the centroid of the area used to estimate the PCSA (point load method). That is, the use of 3 dimensional FEA is fully justified as it allows to account for the entire geometry of the skull (or the structure of interest) rather than modelling it as a beam. In addition, the whole area of muscle's attachment can be loaded (as we did, see below) rather than just a point. 3D approaches clearly give more accurate predictions (See Davis et al. 2010; Cox et al., 2016 among many others).

Further, the use of FEA is particularly fitting our experiments as we aimed at modelling the behaviour of theoretical shapes, which can be more accurately simulated using a more complex strategy. In recent years most of the investigations carried on the evolution of trade-offs in anatomical structures widely used FEA (see Polly et al., 2016; Dickson et al., 2019, 2021; Deakin et al., 2022 and many others).

We further note that the use of FEA allows us to quantitatively examine the relationship between bite force and cranial strength.

Wroe, S., McHenry, C., & Thomason, J. (2005). Bite club: comparative bite force in big biting mammals and the prediction of predatory behaviour in fossil taxa. *Proceedings of the Royal Society B: Biological Sciences*, 272(1563), 619-625.

Davis, J. L., Santana, S. E., Dumont, E. R., & Grosse, I. R. (2010). Predicting bite force in mammals: two-dimensional versus three-dimensional lever models. *Journal of Experimental Biology*, 213(11), 1844-1851.

Cox, P. G., Rinderknecht, A. & Blanco, R. E. Predicting bite force and cranial biomechanics in the largest fossil rodent using finite element analysis. *J Anat* **226**, 215–223 (2015).

Polly, P. D. *et al.* Combining geometric morphometrics and finite element analysis with evolutionary modeling: towards a synthesis. *J Vertebr Paleontol* **36**, e1111225 (2016).

Dickson, B. V. & Pierce, S. E. Functional performance of turtle humerus shape across an ecological adaptive landscape. *Evolution (N Y)* **73**, 1265–1277 (2019).

Dickson, B. V., Clack, J. A., Smithson, T. R. & Pierce, S. E. Functional adaptive landscapes predict terrestrial capacity at the origin of limbs. *Nature* **589**, 242–245 (2021).

Deakin, W. J. *et al. Increasing morphological disparity and decreasing optimality for jaw speed and strength during the radiation of jawed vertebrates. Sci. Adv* vol. 8 <https://www.science.org> (2022).

This issue is further exacerbated by the way the biomechanical analyses have been performed, which makes me quite sceptical about their biological accuracy. Again, this is a very important problem here as the downstream analyses are based on the functional variables extracted from those biomechanical models.

First, I do not understand why the authors used the inverse of the von Mises stress (L416) as an estimate of bite force while it can be directly computed as the reaction force at the constrained node on the teeth (L415)? Moreover, I think that the authors' argument to justify this approach ("shape capable of sustaining higher loading conditions should display lower von Mises stress" [L416-417]) is also problematic. This assumes that the cranium is, as a whole unit, optimised (or not) to resist feeding loads (stress value are here averaged across the whole skull [L418]), while we know that mammalian cranium experience marked bone strain (and therefore stress) gradients during biting. Some areas of the cranium such as the bow ridge in primates experience very low strain magnitudes during biting compared to others, suggesting that bone mass in those areas is unrelated to the resistance to feeding loads (i.e. they are overdesigned to resist those loads) (e.g. Godinho *et al.* 2018 *Nature Ecol. Evol.*). Therefore, as the cranium performs many functions, there might not be a single optimality criterion in the design of different regions of the cranium of the same animal, which make the relationship between the loading regime (the combination of forces applied to the skull during biting) and bone distribution in the cranium far more complex than what is presented here (see for discussion Ross *et al.* 2019 *J. Exp. Biol.*; Hylander & Johnson 1997 *Am. J. Biol. Anthropol.*). Therefore, I do not think that the author employed the correct approach to estimate their functional variables, as what really matters for the trade-off studied here is the musculoskeletal function (how muscle forces are generated and transferred) rather than the structural behaviour (how the skeleton resists those loads) of the system.

Answer: The reviewer observes that von Mises stress may not necessarily be a direct indicator of bite force. We accept this and have further rerun analyses based on calculated bite force values taken directly from the models.

We performed comparisons and repeated all the main analyses presented in this study using the bite forces estimated from our FEA models and included these in the Supplementary Information. To this aim, we performed the following steps:

1) The bite force values were compared to those obtained from different studies incorporating our same species (i.e. Christensen & Adolfsen, 2005; Attard *et al.*, 2014; Sakamoto *et al.* 2019; Hartstone-Rose *et al.*, 2019) and we included this information in Supplementary Table 2.

2) We computed the correlation between von Mises stress values and bite force values. Results indicated a strong correlation between the two metrics (r Pearson = 0.82, p -values < 0.001; ρ Spearman = 0.88; p -value < 0.001), strongly suggesting that downstream analyses should be very similar regardless of the metric applied.

3) Using bite force values instead of von Mises stress, we estimated the best fitting interpolation strategy (a second degree TPS showed the lowest RMSE) and produced a performance surface. This performance surface for bite force directly estimated from the models (reproduced here and included as the new Supplementary Figure 1) is, indeed, very similar to the one in figure 1c (see below).

4) Following the scaling of the estimated bite force values we computed new trade-off weight w values based on bite force values and used them to repeat downstream analyses.

5) We computed evolutionary rates for bite force values using the same approaches (Bayesian estimates and Ridge Regression) described in the Methods section and computed correlations between shape evolutionary rates and trade-off weight w values (based on bite force values) evolutionary rates. Results showed that shape rates were again uncorrelated with the weight w rates based on bite force values (r Pearson = 0.026, p -value = 0.24; ρ Spearman = 0.013, p -value = 0.81). The same holds when using phylogenetically corrected tip rates values (r Pearson = 0.022, p -value = 0.81; ρ Spearman = 0.12, p -value = 0.16).

As the reviewer will note, the distribution of rates of the trade-off weight w based on bite force values (see pictures below and the new Supplementary Figure 2) is highly comparable to the distribution of rates of weight w based on VM stress as displayed in figure 2b.

To further assess this, we computed correlations between the trade-off weight w tip rates based on von Mises stress and w tip rates based on bite force values. Results showed that tip rates of both trade-off weights w were significantly correlated (r Pearson = 0.71, p -value < 0.001; ρ Spearman = 0.83, p -value < 0.83). The same holds when using phylogenetically corrected tip rates values (r Pearson = 0.67, p -value < 0.001; ρ Spearman = 0.72, p -value < 0.73).

6) We repeated our analysis of the relationship between the force-velocity trade-off and morphological disparity. Again, we used the same approach described in the main text, but using the trade-off weight w based on bite force values. The distributions of weight w volume and disparity showed again an overlapping pattern, with three identifiable peaks (see picture below). However, there was some difference in the shape of the peaks when compared to the distribution obtained when using the weight w based on von Mises stress data, in particular the peak around a value of 0.41 which was higher. Nonetheless, morphological disparity and weight w volume were again positively correlated (r Pearson = 0.77, p -value < 0.001; ρ Spearman = 0.58, p -value < 0.001; χ^2 = 0.63, p -value < 0.001).

These results show that the same conclusions can be drawn using direct bite force estimates, or von Mises stress data as a proxy. That is, we are confident that our inferences and main conclusions are not influenced by the metrics chosen as a proxy of the “force” side of the “force-velocity” dichotomy.

However, we must acknowledge that the high correlation between von Mises stress and bite force estimates detected here might not universally apply to other mammalian clades. Hence, we recommend that the metrics chosen as a force proxy should be fully justified in further investigations.

Then, only the temporalis and masseter groups are modelled (L438-440), and at no point in the manuscript do the authors explain why the pterygoid muscle group was not included. I am sorry if I

have missed something, but I do not understand the rationale behind this choice, especially as it likely has consequences on the outputs of the models. Also, there is little information as to how the muscle orientations were reconstructed; for instance, does the temporalis muscle wrap onto the cranial vault or was it modelled as following a straight path? The way muscles are reconstructed indeed affects reaction forces (see for instance Groening et al. 2013 J. Royal Soc. Interface).

Answer: The reviewer noted that we did not model the pterygoid musculature. However, the Pterygoid musculature plays a relatively minor role in jaw adduction among mammalian carnivores and is typically not included in 2 or 3D modelling (see Thomason, 1991; Wroe et al., 2005; Tseng, 2009; Tseng et al., 2011 and many others).

We described the process of muscles modelling and orientation at lines 463-477, which largely follows that applied in previously published studies (see Attard et al., 2014). We modelled each muscle group as truss elements wrapping the approximated areas of origin and insertion, that is (as noted above) we avoided “point loading” our models. Orientation has been guided using the mandible to determine the direction of each truss element.

Moreover, this is also very not clear to me how the authors estimated muscle force in the 49 ‘real’ specimens (L432-469). This is quite problematic as muscle morphology and architecture (muscle volume, fibre lengths, pennation angle) are important here; for instance, increasing muscle size (with fibre length unchanged) or decreasing fibre length (with muscle size unchanged) while maintaining the same lever arms would result in greater bite force. Moreover, there is no information on the gape at which maximal bite force was estimated, while we know that the configuration of the muscles and the length-tension relationship of sarcomeres impact the gape at which maximal bite force is generated. I am therefore quite concerned that we miss quite a lot here about the musculoskeletal function, which eventually skews the conclusions about functional trade-offs between bite force and jaw-closing speed.

Answer: In carnivores optimal bite is generally achieved at a 35° gape (see Attard et al., 2014, among others). That is, our simulations were all performed at that angle and this information has been included in the text (lines 479-480).

Regarding muscle forces, here we specify that the aim of our analyses is comparative and not validative. Our main goal is to generate reliable simulation of theoretical shapes, including hypothetical geometries not represented by extant or extinct taxa. We used 49 real specimens to measure the capacity of the theoretical shapes to predict the behaviour of real specimens when mapped onto the functional landscape. Nonetheless, our sample included extinct species such as the cave bear *Ursus spelaeus*, the marsupial lion *Thylacoleo carnifex* and the marsupial ‘wolf’ *Thylacinus cynocephalus*, for these species direct measurements of muscle morphology and architecture are not available. Hence, they must be estimated using other approaches such as the dry skull method, which has provided the stronger approximation of muscle size and PCSA (Dickinson et al., 2021).

Furthermore, as detailed in lines 482-484, we scaled our models to the same surface area to remove the effect of size according to standard protocols in comparative analyses (see Dumont et al., 2011). That is, the same force has been applied to all the models (see Tsang et al., 2019, among others). In this scenario, differences in performance outputs are due to morphological variation only. This is highly recommended when comparing theoretical models which are generated to explore the performance of shapes populating areas of the morphospace not covered by real specimens (see Polly et al., 2016; Dickson et al., 2019; Dickson et al., 2021; Sansalone et al., 2022).

For taxa where published data are available, I would have appreciated to see how the relative differences in bite force between taxa predicted with FEA reflects those observed with in vivo bite force measurements. This would have allowed the authors to demonstrate the biological accuracy of their 49 FE models, which were used as a reference to estimate the accuracy of the FEAs made on the functional surfaces (L194-197).

Answer: In vivo bite force data are available only for a handful of carnivores species. We further note that unless the few individuals for which bite force data are available are of the same size as those in our sample, it would be difficult to draw meaningful comparisons.

Moreover, unless individuals have been sedated and their jaw muscles externally stimulated, it is impossible to assess whether they exerted maximal bite forces (see Ellis et al. 2008).

We reiterate that our predictions of bite force and stress have been provided in an entirely comparative context, as is standard in every comparative study (see Polly et al., 2016; Tsang et al., 2019; Tseng, 2013; Marcè-Noguè et al., 2017).

For reasons detailed above, I have very little confidence in the biomechanical models presented here, and, as a consequence, in the functional variables underpinning the conclusions on the morpho-functional evolution of the carnivore feeding apparatus. Unfortunately, and despite the interesting question addressed and the scope of the study, I cannot support this article for publication in Nature Communications.

Answer: We hope that the reviewer will appreciate the effort we have made to address his/her concerns and will concur that the extensive additional analyses we have undertaken address these and further support the robustness of our results.

Reviewers' Comments:

Reviewer #1:

Remarks to the Author:

I appreciate that the authors have considered my comments on the previous manuscript. The authors have adequately addressed the issues I suggested in the previous review. The new Supplementary Table 2 is very informative.

Please note that the reference that should have been placed after the word "in" on Line 411 is not shown in the PDF of the current version of the manuscript.

Reviewer #2:

Remarks to the Author:

I think that the authors have provided all the information to respond to my comments. In my opinion, the authors have made modifications to the manuscript that correctly addressed the things mentioned by me and the other reviewers. Accordingly, I recommend the manuscript publication in Nature Communications.

Reviewer #3:

Remarks to the Author:

Dear Authors and Editors

This new version of the manuscript of Sansalone et al. has been improved, and I appreciate the effort that the authors have made to answer my comments. Please find below some additional comments on this revised version.

The authors performed the analyses with bite force values instead of von Mises stress values. I suggest including a few lines justifying the choice of the inverse of von Mises over the reaction force at the bite point (i.e. bite force) (P9L187 or P20L431). Moreover, it is important to clearly state in the M&Ms that muscle forces used to calculate bite force are estimated based on the dry skull method (P21L450). Therefore, the strong correlation between the bite force estimates of this study and previous ones (supp data P3L64 and supp table S2) is not surprising, as they all employed the dry skull method for muscle force estimation. It is worth noting that other studies have pointed out biases in this modelling approach compared with dissection-based musculoskeletal models (e.g. Penrose et al. 2019 Anatomical Record), something that should be briefly discussed in my opinion.

Quantitative dissections of carnivores indicate variability in the PCSA of the medial pterygoid relative to other adductors, as well as jaw size (e.g. Ito & Endo 2016 Mammal Study). I suggest the authors briefly discuss their choice not to model the medial pterygoid in the context of the results of quantitative dissection available in the literature.

The authors claim that the optimal gape for all carnivores is 35 degrees (P22L480). To my knowledge, this statement is not supported by the anatomical and physiological data available. Attard et al. 2014, which is cited to support this claim L480, do not give any evidence for it –so citing this study in this context is misleading. The optimal gape angle is probably quite variable among carnivores because of the diversity in the geometry of the musculoskeletal system and muscle architecture, which might be in turn linked to the variation in the relative size of the preys (as the authors note L320).

Contrary the authors' claim in their answer to one of my previous comment, a static model of the musculoskeletal system: (1) is not necessarily a 2D model; (2) does not necessarily require the muscle force vectors to be directed toward the centroid of the of attachment area. Static or dynamic models can accurately capture the 3D muscle morphology and mechanics by dividing muscles into several lines of action (e.g. Curtis et al. 2008 Anatomical Record) and using more or less complex mathematical models for their physiology. Therefore, I maintain the view that FEM may not be the most appropriate approach to test the interesting hypotheses about the force/velocity trade-off in the musculoskeletal system. We are here dealing with force generation and transmission, rather than

structural resistance of the skeleton to the loading regime. Therefore, I would recommend the authors to better justify why they choose to use FEM, which entails more ad hoc hypotheses than a static musculoskeletal model.

REVIEWERS' COMMENTS

Reviewer #1 (Remarks to the Author):

I appreciate that the authors have considered my comments on the previous manuscript. The authors have adequately addressed the issues I suggested in the previous review. The new Supplementary Table 2 is very informative.

Please note that the reference that should have been placed after the word “in” on Line 411 is not shown in the PDF of the current version of the manuscript.

Answer: We are glad the Reviewer appreciated our efforts to improve the manuscript quality. We added the missing reference in line 411 (it was ref. #53).

Reviewer #2 (Remarks to the Author):

I think that the authors have provided all the information to respond to my comments. In my opinion, the authors have made modifications to the manuscript that correctly addressed the things mentioned by me and the other reviewers. Accordingly, I recommend the manuscript publication in Nature Communications.

Answer: We are happy that the Reviewer has found the changes we made to the last version satisfactory and we thank him/her for recommending the acceptance of our manuscript.

Reviewer #3 (Remarks to the Author):

Dear Authors and Editors

This new version of the manuscript of Sansalone et al. has been improved, and I appreciate the effort that the authors have made to answer my comments. Please find below some additional comments on this revised version.

Answer: We are glad that the Reviewer appreciated our efforts. As it will become apparent in the next answer to the Reviewer's comments, we have now made an additional effort to further accommodating his/her suggestions.

The authors performed the analyses with bite force values instead of von Mises stress values. I suggest including a few lines justifying the choice of the inverse of von Mises over the reaction force at the bite point (i.e. bite force) (P9L187 or P20L431).

Answer: We added a few lines in Supplementary Information to detail the choice of using von Mises stress data rather than reaction forces. Basically, von Mises stress is obtained over the entire geometry of the skull rather than at a single point, which in turns makes this kind of data a better choice in a study aimed at characterising the relationship between trade-off in function and overall cranial morphology.

Moreover, it is important to clearly state in the M&Ms that muscle forces used to calculate bite force are estimated based on the dry skull method (P21L450). Therefore, the strong correlation between the bite force estimates of this study and previous ones (supp data P3L64 and supp table S2) is not surprising, as they all employed the dry skull method for muscle force estimation. It is worth noting that other studies have pointed out biases in this modelling approach compared with dissection-based musculoskeletal models (e.g. Penrose et al. 2019 Anatomical Record), something that should be briefly discussed in my opinion.

Answer: Thank you for the suggestion, we added a short explanation in the Supplementary Information and acknowledged the existence of potential biases. However, specifically on the study by Penrose and colleagues that the Reviewer is mentioning, it should be noticed that: 1. This study compares two methods of estimation rather than comparing methods of

estimation with actual (*in vivo*) measurements so, even if there were differences in the values (a bias) among estimation methods, this would not be informative with respect to differences between estimates and true values. 2. This study does not provide strong evidence for substantial differences between methods which are, in fact, at best weak. Indeed, direct quotes from the study by Penrose and colleagues include “there was weak evidence of a possible skew [...] However, the slopes in both cases were not significantly different from 1.” or “dry skull derived values are a good approximation of RPCSA values where dissection derived data is unobtainable”. 3. Even if there were a bias between our method of estimation of choice (the “dry skull” method) and the true, relevant, values (i.e., *in vivo*), as a bias is a consistent difference in means (i.e., a variation in location of the distribution of values), this would not affect studies such as ours which focus on variation among species (i.e., among values in the distribution) but only studies combining estimates obtained using different methods. This is particularly true for our study, considering its relatively large phylogenetic scale of investigation. 4. If we had used a different estimation method, the estimates on extinct taxa we obtained would have been precluded.

To sum up, we follow the Reviewer’s suggestion of mentioning the potential for bias, but we do not think – given the current state of knowledge in the field – that this is particularly problematic for our analysis.

Quantitative dissections of carnivores indicate variability in the PCSA of the medial pterygoid relative to other adductors, as well as jaw size (e.g. Ito & Endo 2016 Mammal Study). I suggest the authors briefly discuss their choice not to model the medial pterygoid in the context of the results of quantitative dissection available in the literature.

Answer: We did not model the pterygoid muscle at its origin and insertion points were not obvious in the warped theoretical models. This could have, potentially, led to incorrect estimates of the stress values over the cranium. We reported it in the text for the sake of clarity.

The authors claim that the optimal gape for all carnivores is 35 degrees (P22L480). To my knowledge, this statement is not supported by the anatomical and physiological data available. Attard et al. 2014, which is cited to support this claim L480, do not give any evidence for it –so citing this study in this context is misleading. The optimal gape angle is probably quite variable among carnivores because of the diversity in the geometry of the musculoskeletal system and muscle architecture, which might be in turn linked to the variation in the relative size of the preys (as the authors note L320).

Answer: We thank the Reviewer for noticing this oversight, which we now promptly corrected (line 480). Indeed, gape is variable among carnivores (but optimal gape is unknown for many of the species in our analysis). We set the angle at 35° for each simulation to obtain estimates that could be comparable across species (as well as theoretical models, for which optimal gape is by definition unknown). We also removed the reference, as we agree with the Reviewer it was included erroneously.

Contrary the authors’ claim in their answer to one of my previous comment, a static model of the musculoskeletal system: (1) is not necessarily a 2D model; (2) does not necessarily require the muscle force vectors to be directed toward the centroid of the of attachment area. Static or dynamic models can accurately capture the 3D muscle morphology and mechanics by dividing muscles into several lines of action (e.g. Curtis et al. 2008 Anatomical Record) and using more or less complex mathematical models for their physiology. Therefore, I maintain the view that FEM

may not be the most appropriate approach to test the interesting hypotheses about the force/velocity trade-off in the musculoskeletal system. We are here dealing with force generation and transmission, rather than structural resistance of the skeleton to the loading regime. Therefore, I would recommend the authors to better justify why they choose to use FEM, which entails more ad hoc hypotheses than a static musculoskeletal model.

Answer: We totally agree with the reviewer that accurate modelling of musculoskeletal systems may not necessarily involve the use of FEM. And this would have been indeed a correct observation if our only aim had been to “accurately capture 3D muscle morphology and mechanics”. However, we want to stress that the use of FEA is fully justified in our study, where the main focus is on the evolution of cranial shape diversity in response to the force/velocity trade-off. Indeed, contrary to the Reviewer’s claim, we are not “dealing with force generation and transmission” on their own, but the relationship between the mechanical properties of skulls and their shape. In this context, using methods that take into account not only muscle morphology but also information on bony structures (skull) represents the optimal choice.